# Hyperparameter Transfer Laws for Non-Recurrent Multi-Path Neural Networks

Haosong Zhang [* 1]   Shenxi Wu [* 1]   Xingjian Ma [1]   Shirui Bian [1]   Yichi Zhang [2]   Xi Chen [2]   Wei Lin [1]

## Abstract

Deeper modern architectures are costly to tune, and the base learning rate is often one of the most sensitive hyperparameters. Maximal Update Parametrization ($\mu$P) helps explain why many hyperparameters transfer across width. Yet depthwise learning-rate scaling is less understood for modern architectures with convolution, residual aggregation, and attention. To unify various non-recurrent multi-path neural networks such as CNNs, ResNets, and Transformers, we introduce an architecture-dependent notion of effective depth. Under stabilizing initializations and a maximal-update criterion, we derive a shared leading-order -3/2 law for the base learning-rate scale as effective depth grows. Here, the budget controls typical one-step representation-update energy at initialization, and effective depth counts sequential update-bearing units while absorbing fixed local structure into constants. Experiments across diverse architectures confirm the predicted slope and enable reliable zero-shot transfer of learning rates across depths and widths, turning depth scaling into a predictable hyperparameter-transfer problem.

## 1. Introduction

In the process of scaling deep learning, training cost grows not only with the number of parameters and data(Hoffmann et al., 2022), but also because hyperparameter tuning is expensive and remains largely experience-driven in practice(Cohen et al., 2021; Hayou & Yang, 2023; Godbole et al., 2023; Kalra & Barkeshli, 2024). Hyperparameters,

including learning rate, regularization, initialization, etc., often require repeated trials at the target scale. Nowadays, this step has become a non-negligible computational and engineering bottleneck. Recently, an important idea is "tuning" on a smaller proxy model and transfer. Under a suitable parameterization, one can identify effective hyperparameters on a small model and transfer them zero-shot to the large one, which significantly reduces the overall tuning cost.

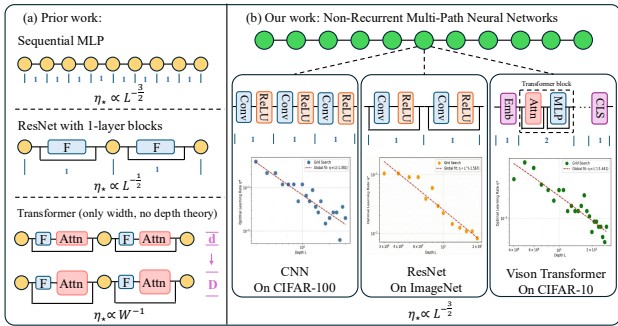

*Figure 1.* Overview of prior work and our results on depth-wise learning-rate scaling. 1 denotes one depth unit, d and D denote width. (a) Prior work mainly analyzes sequential networks or special cases, and Transformers are often discussed only under width scaling. (b) We treat CNNs, ResNets, and Transformers as non-recurrent multi-path networks and obtain a unified depth law for the maximal-update learning rate.

The key to this paradigm is to find a parameterization under which training dynamics remain comparable as the model scale changes. Yang et al. (2021) showed that Maximal Update Parametrization, denoted $\mu$P, aligns update scales under width scaling, motivating the zero-shot hyperparameter transfer paradigm $\mu$Transfer, where one tunes on a narrow model and transfers to a wider one. However, changing depth introduces more complex scaling effects than those from width scaling. Even in sequential networks like ReLU MLPs, existing analysis suggests that the $\mu$P maximal-update learning rate depends on depth $L$ and scales as $L^{-3/2}$ (Jelassi et al., 2023). Modern high-performance models have more complex connection patterns such as spatial convolution and weight sharing in CNNs, residual connections in ResNets, attention modules and residual branches in Transformers. For these models, a unified depthwise learning-rate scale is still missing. Without such a scale, changing depth still requires costly learning-rate

[*]Equal contribution   [1]Fudan University, Shanghai, China [2]New York University, New York, NY, USA. Correspondence to: Yichi Zhang <zhangyichi@stern.nyu.edu>, Xi Chen <xc13@stern.nyu.edu>, Wei Lin <wlin@fudan.edu.cn>.

*Proceedings of the 43$^{rd}$ International Conference on Machine Learning*, Seoul, South Korea. PMLR 306, 2026. Copyright 2026 by the author(s).

re-tuning.

Unlike recurrent architectures such as RNNs (Elman, 1990; Werbos, 2002), LSTMs (Hochreiter & Schmidhuber, 1997) and GRU (Cho et al., 2014), these models are non-recurrent. They can be viewed as feedforward computation graphs with multiple parallel paths and branch aggregation. This multi-path view is especially explicit in residual networks (He et al., 2016; Veit et al., 2016) and recent work has shown cross-depth transfer in specific settings. For example, **?** combine a $1/\sqrt{\text{depth}}$ residual-branch scaling with $\mu$P, enabling hyperparameter transfer across both width and depth in convolutional ResNets and Vision Transformers(ViTs). Meanwhile, Yang et al. (2023) propose Depth-$\mu$P and further points out that existing infinite-depth analyses would face new challenges when the block structure is deeper, for example, in Transformers (Vaswani et al., 2017; Wang et al., 2019). Thus, the main idea of this paper is to develop a unified vision of depth scaling for non-recurrent structures.

This paper unifies CNNs, ResNets, and Transformers as parallel-structured non-recurrent architectures and studies how pre-activation update scale is allocated under parallel aggregation.(Fig. 1). [1] Under this framework, we derive a shared leading-order -3/2 law for the base learning-rate scale. Operationally, the rule calibrates a learning rate at a reference depth and rescales it to a new effective depth. Theoretically, we develop a depth-scaling framework for parallel-structured networks that absorbs the maximal-update idea of $\mu$P and extends sequential-MLP depth–LR arguments to CNNs, ResNets, and Transformers. Empirically, we build an automated learning-rate search and fitting pipeline to evaluate the finite-depth law across architectures, datasets, and training variants, with additional checks for proxy widths, later epochs, and direct zero-shot depth transfer. We summarize the resulting transfer guidance in Table 3.

**Main contributions.**

- We introduce the concept of effective depth unit for non-recurrent multi-path networks, including sequential MLPs, CNNs as well as ResNets, and Transformers under architecture-specific conventions.

- We generalize the maximal-update principle of $\mu$P to heterogeneous multi-path graphs via a global update-energy budget, yielding a practical parameterization

---

[1]For a scaling variable $s$, $f \propto g$ means $f(s)/g(s) \to \kappa \in (0,\infty)$, i.e., $f(s) = \kappa\, g(s)\,(1 + o(1))$ with $\kappa$ independent of $s$; hence $f = \Theta(g)$ (not conversely). Operationally, we use this as a transfer rule by calibrating $\kappa$ at a reference scale and extrapolating with the power-law exponent. Our theory yields the proportional form for MLP/CNN; otherwise we state the more conservative $\Theta(\cdot)$ result. See Sec. 3.4 for the definition of $\Theta(\cdot)$.

and a learning-rate scale under parallel aggregation.

- Under stabilizing initializations and the network-wide criterion, we prove a unified depth–LR power law with exponent $-3/2$ for CNNs, ResNets, and Transformers, enabling a zero-shot cross-depth transfer rule.

- We build an automated learning-rate search and fitting pipeline and validate the law across architectures, datasets, and training variants; we summarize transfer guidance and its boundaries in Table 1 and Table 3.

**Organization.** Sec. 2 reviews related work. Sec. 3 introduces our framework and theoretical analysis for depth-aware learning-rate scaling. Sec. 4 presents the automated learning-rate search pipeline and experiments. Sec. 5 concludes, and Sec. 6 discusses limitations and future work. Appendices show proofs and additional results.

## 2. Related Works

**Initialization and signal propagation.** Training very deep networks often requires controlling signal and gradient scales at initialization. Classic fan-in scaling schemes such as those of Glorot & Bengio (2010) and He et al. (2015) are designed to keep activations and gradients from vanishing or exploding in deep feedforward networks. Mean-field analyses further characterize trainability through depth scales and dynamical isometry conditions (Schoenholz et al., 2016; Xiao et al., 2018). For convolutional and residual architectures, several works study how initialization or residual-branch scaling should depend on depth, including analyses of deep ResNets (Zagoruyko & Komodakis, 2016; Xie et al., 2017; Taki, 2017), Stable ResNet and depth-dependent residual scaling for stable signal propagation (**?**), and normalization-free variants such as Fixup (Zhang et al., 2019), SkipInit (De & Smith, 2020), and ReZero (Bachlechner et al., 2021). In Transformers, stability is also closely tied to normalization and residual scaling. This has motivated both theoretical and practical adjustments, such as analyzing LayerNorm placement (Xiong et al., 2020) and proposing DeepNorm with derived initialization for very deep Transformers (Wang et al., 2024) . We adopt these widely used initialization and scaling principles throughout, and we build our theoretical framework on top of them; see Sec. 3.

**Hyperparameter transfer and maximal-update parameterizations.** The cost of tuning large models has motivated work on transferring hyperparameters from small proxy models. A prominent theoretical approach is Maximal Update Parametrization, or $\mu$P, introduced in Tensor Programs V by Yang et al. (2021). By aligning update scales under width scaling, $\mu$P makes many training-critical hyperparameters stable across widths, enabling the

zero-shot transfer paradigm $\mu$Transfer. Recent theory has begun to formalize this width-transfer principle: Hayou (2026) proves learning-rate transfer under $\mu$P in width-scaled linear MLPs, showing that the optimal learning rate converges as width grows, while analogous transfer can fail under standard or NTK parameterizations. These width-wise results are complementary to our setting, where the scaling variable is effective depth rather than width. Practical tooling and recipes include the open-source `mup` package (Microsoft Research, 2022a) and Transformer-oriented $\mu$P parameterizations released in accompanying codebases (Microsoft Research, 2022b). Complementary viewpoints connect hyperparameter scales to feature-learning dynamics and offer a more flexible lens on transfer across regimes (Chizat & Netrapalli, 2024).

Extending transfer laws from width to depth is more subtle (Hestness et al., 2017; Kaplan et al., 2020), since changing depth can alter effective path counts, normalization statistics, and residual accumulation. In the sequential ReLU MLP setting, Jelassi et al. (2023) show that the maximal-update learning rate depends nontrivially on depth and scales as $L^{-3/2}$. Architecture-aware analyses provide another route by deriving topology-dependent maximal learning-rate prescriptions for general computation graphs, such as PathSum-based rules (Chen et al., 2024). For residual architectures, the $1/\sqrt{\text{depth}}$ residual-branch scaling used in depth-transfer methods is closely connected to the stability perspective of Stable ResNet (Hayou et al., 2021). Building on depth-normalized residual scaling, Bordelon et al. (2023) empirically demonstrate cross-depth and cross-width hyperparameter transfer in residual architectures, including convolutional ResNets and Vision Transformers, and motivate the behavior using dynamical mean-field descriptions. From the Tensor Programs perspective, Yang et al. (2023) propose Depth-$\mu$P and discuss challenges that arise for infinite-depth feature learning when block structures become more complex. Recent MoE-Transformer work studies a different but related transfer problem, where width, depth, number of experts, and expert hidden size are varied jointly under MoE-specific parameterizations (Jiang et al., 2026). Minimal models, such as the deep linear analysis of Bordelon & Pehlevan (2025), further clarify how data, width, depth, and parameterization interact behind hyperparameter transfer. Overall, these works show that depthwise transfer can be achieved in several important settings. We build on this line by combining an effective-depth convention with a network-wide maximal-update budget to obtain a compact depthwise learning-rate scale for CNNs, ResNets, and Transformers.

**Non-recurrent multi-path architectures and depth scaling.** Many modern vision and language models are non-recurrent feedforward computation graphs with multiple parallel paths and branch aggregation. Residual networks make this structure explicit through skip connections (He et al., 2016; Balduzzi et al., 2017), and can be interpreted as ensembles over paths of different lengths (Veit et al., 2016). Mean-field studies of randomly initialized ResNets further analyze how depth interacts with stability and signal propagation (Yang & Schoenholz, 2017). Transformers also follow a residual multi-branch pattern that alternates attention and feedforward sublayers (Vaswani et al., 2017). This multi-path viewpoint motivates the dense non-recurrent architecture class used in our effective-depth convention.

**Automated hyperparameter optimization and proxy training.** Hyperparameter optimization has a long history, with methods ranging from Bayesian optimization (Snoek et al., 2012) to multi-fidelity and early-stopping strategies such as Hyperband (Li et al., 2018). At large scale, system efforts such as Hydro use surrogate models and scaling techniques to reduce end-to-end tuning cost in clusters (Hu et al., 2023). Learning-rate schedules and warmup are themselves important hyperparameters, and recent work studies when warmup is needed and how it can be reduced in GPT training (Kosson et al., 2024). Our empirical pipeline uses automated learning-rate search as a calibration and evaluation tool for the proposed depthwise learning-rate scale across architectures.

## 3. Methods

This section establishes the common setup shared by our theoretical analysis and experiments. We first introduce architecture-dependent depth units, notation, and the initialization conventions used throughout (Sec. 3.1). We then define our network-wide maximal-update criterion, Arithmetic-Mean $\mu$P, which provides a single width-robust learning-rate scale for heterogeneous multi-path architectures (Sec. 3.2). Finally, under this unified framework we derive depthwise learning-rate scaling laws for CNNs, ResNets, and Transformers, and discuss when architectural details only affect constant factors (Secs. 3.3–3.5).

Throughout this section, $\eta^\star$ denotes the base learning-rate scale selected by the $\mu$P update budget. This is the quantity that will later be calibrated empirically by early-training learning-rate sweeps in Sec. 4.

### 3.1. Initialization and architectural conventions

**Depth units and notation.** We study non-recurrent feedforward models and use a unified depth index $\ell$ to denote depth units, i.e., the units with respect to which we state depthwise learning-rate scaling laws. The mapping from implementation details to depth units is architecture-dependent, but follows a common convention: we count sequential update-bearing units along the input–output back-

bone after contracting fixed local branch templates. A plain affine or convolutional transform followed by a pointwise nonlinearity contributes one depth unit, and each residual aggregation/update contributes one depth unit. Fixed $O(1)$ internal structure inside a branch or block is absorbed into the local template. Operations such as reshaping, pooling, concatenation, and normalization placement do not by themselves create new depth units when their Jacobians remain bounded at initialization; they affect constants in the update budget rather than the leading depth exponent.

**Fan-in and activation statistics.** For a weight tensor $W$, we denote by $\mathrm{fan}_{\mathrm{in}}(W)$ the number of inputs contributing to one output. For a linear layer $W \in \mathbb{R}^{d_{\mathrm{out}} \times d_{\mathrm{in}}}$, we have $\mathrm{fan}_{\mathrm{in}}(W) = d_{\mathrm{in}}$. For a convolution with kernel support $\mathcal{K} \subset \mathbb{Z}^d$ of size $|\mathcal{K}| = k$ and $C_{\mathrm{in}}$ input channels, $\mathrm{fan}_{\mathrm{in}}(W) = k\,C_{\mathrm{in}}$. Throughout, $\sigma(\cdot)$ denotes a pointwise activation function. We also define the gating factor $q := \mathbb{E}[\sigma'(Z)^2]$ for $Z \sim \mathcal{N}(0,1)$. In our theoretical proofs we use ReLU, in which case $q = 1/2$. Under the mean-field normalization where pre-activations are $O(1)$ at initialization, fan-in initialization can be written as

$$W_{ij} \sim \mathcal{N}\left(0,\ \frac{1}{q\,\mathrm{fan}_{\mathrm{in}}(W)}\right), \qquad (1)$$

which reduces to the standard He initialization $\mathrm{Var}(W_{ij}) = 2/\mathrm{fan}_{\mathrm{in}}(W)$ for ReLU. The initialization will be used in this paper and serves as the basis for our framework.

**MLPs and homogeneous CNNs.** For sequential models, we write $z^{(0)}(x) = x$ and

$$z^{(\ell)}(x) = W^{(\ell)}\,\sigma\left(z^{(\ell-1)}(x)\right), \qquad \ell = 1,\dots,L, \quad (2)$$

where $L$ is the number of depth units.

For homogeneous CNNs[2], $W^{(\ell)}$ denotes a convolutional kernel followed by a pointwise nonlinearity. At depth $\ell$, the feature map has $C_\ell$ output channels indexed by $j \in \{1,\dots,C_\ell\}$, and a spatial index set $\Lambda_\ell \subset \mathbb{Z}^d$ collecting all spatial locations. For example, in 1D we may take $\Lambda_\ell = \{0,1,\dots,N_\ell - 1\}$, while in 2D we may take $\Lambda_\ell = \{0,\dots,H_\ell-1\} \times \{0,\dots,W_\ell-1\}$ with $N_\ell := |\Lambda_\ell| = H_\ell W_\ell$. The convolution at location $p \in \Lambda_\ell$ aggregates inputs from locations $p + \Delta$, where $\Delta$ ranges over a kernel offset set $\mathcal{K}_\ell \subset \mathbb{Z}^d$ (e.g., for a $3 \times 3$ kernel, $\mathcal{K}_\ell = \{-1,0,1\}^2$ and $k_\ell := |\mathcal{K}_\ell| = 9$). With circular padding, $p + \Delta$ is interpreted modulo the spatial grid (torus indexing), so boundary effects are absent in the idealized setting. In our main derivations, we first consider the homogeneous case where $(C_\ell, \Lambda_\ell, \mathcal{K}_\ell)$ do not vary with $\ell$; we keep the subscripts for compatibility with variants.

---

[2]Terminology: "homogeneous" means depth-stationary architectural statistics, not positive homogeneity / scale equivariance.

With stride 1, for $\ell = 1,\dots,L$, we write $z^{(0)}(x) = x$ and the pre-activation recursion as

$$z^{(\ell)}_{j,p}(x) = \sum_{i=1}^{C_{\ell-1}} \sum_{\Delta \in \mathcal{K}_\ell} W^{(\ell)}_{j,i,\Delta}\,\sigma\left(z^{(\ell-1)}_{i,p+\Delta}(x)\right). \qquad (3)$$

where $p \in \Lambda_\ell$ and $j \in \{1,\dots,C_\ell\}$. The classifier is global average pooling followed by a linear head, which uses linear fan-in initialization.

**Residual networks.** We index depth at residual-block boundaries. Let $z^{(0)}(x) = x$ and for $\ell = 1,\dots,K$,

$$z^{(\ell)}(x) = z^{(\ell-1)}(x) + F_\ell\left(z^{(\ell-1)}(x)\right), \qquad (4)$$

where $F_\ell$ is the residual branch, whose internal architecture is fixed as $K$ grows. We count the residual addition itself as one depth unit and absorb the fixed $O(1)$ internal structure of $F_\ell$ into the residual-block template. Thus $K$ denotes the number of residual update units along the minimal path, while the internal branch design affects constants in the scaling law. For the weights on residual branches, we use a depth-scaled fan-in initialization inspired by stability analyses of deep residual networks and Stable ResNet-style residual scaling (Taki, 2017; **?**): each residual-branch weight tensor with fan-in $n_{\mathrm{in}}$ is initialized as

$$W_{\mathrm{res}} \sim \mathcal{N}\left(0,\ \frac{1}{q\,K\,n_{\mathrm{in}}}\right), \qquad (5)$$

equivalently applying fan-in initialization and scaling residual-branch weights by $K^{-1/2}$. Weights outside residual branches (e.g., stem/head) use the standard fan-in rule.

**Transformers.** We consider standard Transformer architectures, including both language Transformers and Vision Transformers as special cases, and covering both pre-norm and post-norm variants. Let $z^{(0)}(x)$ denote the token representations produced by the embedding stem (e.g., token embedding in language models, or patch projection in vision models). A Transformer block contains two *sequential* residual updates: a self-attention branch followed by a position-wise feedforward branch. For concreteness, we write the update in the pre-norm form:

$$z \leftarrow z + \mathrm{Attn}(\mathrm{LN}(z)), \qquad z \leftarrow z + \mathrm{FFN}(\mathrm{LN}(z)),$$

where LN is layer normalization, $\mathrm{Attn}$ denotes multi-head self-attention, and FFN denotes the feedforward subnetwork. The post-norm variant is obtained by moving LN after each residual addition. Accordingly, each Transformer block contributes two depth units, one for the attention residual update and one for the FFN residual update. If a Transformer has $D$ blocks and fixed stem/head components, then $L_{\mathrm{tr}} = 2D + O(1)$. In our experiments we use a consistent counting convention for the $O(1)$ stem/head

units; changing this constant-level convention only changes finite-depth offsets and does not affect the leading exponent.

We initialize linear projections in patch embedding, the attention projections, and MLP layers using fan-in initialization. Positional embeddings and the class token are initialized with a small-variance truncated normal distribution, and LayerNorm parameters use unit scale and zero bias.

### 3.2. Maximal-update parameterizations and the arithmetic-mean criterion

**One-step pre-activation updates.** Let $\theta$ denote all trainable parameters and let $\mathcal{L}_{\mathcal{B}}(\theta)$ be the training loss on a mini-batch $\mathcal{B}$. A single (S)GD step with base learning rate $\eta$ updates

$$\theta^+ \;=\; \theta \;-\; \eta \, \nabla_\theta \mathcal{L}_{\mathcal{B}}(\theta). \tag{6}$$

For an input $x$ and depth unit $\ell$ (as defined in Sec. 3.1), we write $z^{(\ell)}(x;\theta)$ for the layer-$\ell$ pre-activations. We use $z_i^{(\ell)}(x;\theta)$ to denote a scalar coordinate of $z^{(\ell)}(x;\theta)$, where $i$ indexes units within layer $\ell$ (neurons in MLPs, channel–location units in CNNs, and token–feature coordinates in Transformers). To connect learning-rate scale to representation update magnitudes, we study the *linearized one-step change* of pre-activations at the start of training. Index scalar parameters by $a$ and write $\partial_a := \partial/\partial\theta_a$. Define $\Delta\theta_a := \theta_a^+ - \theta_a$ and

$$\begin{aligned}
\Delta z_i^{(\ell)}(x) \;&:=\; \sum_a \big(\partial_a z_i^{(\ell)}(x;\theta)\big)\,\Delta\theta_a \\
&=\; -\eta \sum_a \big(\partial_a z_i^{(\ell)}(x;\theta)\big)\,\big(\partial_a \mathcal{L}_{\mathcal{B}}(\theta)\big).
\end{aligned} \tag{7}$$

It is the starting point of maximal-update analyses (Yang et al., 2021; Jelassi et al., 2023). All quantities above will be evaluated at initialization unless stated otherwise.

**Classical maximal-update learning rates.** The maximal-update heuristic chooses a learning rate such that the one-step pre-activation change has an $O(1)$ scale. Concretely, we define the layerwise second moment

$$S_\ell(\eta) \;:=\; \mathbb{E}\Big[\big(\Delta_B z_i^{(\ell)}(x)\big)^2\Big], \tag{8}$$

where the expectation is over random initialization and the sampling of $(x, B)$. Under our symmetric random initialization, $S_\ell(\eta)$ does not depend on the coordinate choice $i$. In the original $\mu$P setting, one sets a reference-layer constraint such as $S_\ell(\eta) = 1$ for a typical hidden layer (or equivalently, for all hidden layers in homogeneous models), which yields a width-robust learning-rate scale and supports zero-shot width transfer of training-critical hyperparameters, including the base learning rate (Yang et al., 2021).

**A network-wide update budget for multi-path architectures.** For non-recurrent multi-path architectures, layer statistics are not homogeneous. Residual additions and branch aggregation can make $S_\ell(\eta)$ vary with depth, so enforcing an identical per-layer constraint can be unnecessarily restrictive or ill-posed. This viewpoint is aligned with architecture-aware analyses of maximal learning rates that explicitly depend on network topology (Chen et al., 2024).

**Arithmetic-mean $\mu$P.** We therefore use a network-wide average update budget. Let $L$ denote the number of depth units under our conventions (Sec. 3.1). We define the network-wide average second moment

$$\bar{S}(\eta) \;:=\; \frac{1}{L}\sum_{\ell=1}^{L} S_\ell(\eta), \; S_\ell(\eta) := \mathbb{E}\Big[\big(\Delta_B z_i^{(\ell)}(x)\big)^2\Big]. \tag{9}$$

We say the network is in the arithmetic-mean $\mu$P (AM-$\mu$P) regime if $\bar{S}(\eta) = 1$, and define the AM-$\mu$P learning-rate scale $\eta^\star$ by $\bar{S}(\eta^\star) = 1$.

This choice has three properties that are important for our setting. First, it reduces to the classical maximal-update criterion in homogeneous models: if $S_\ell(\eta) \approx S(\eta)$ for all $\ell$, then $\bar{S}(\eta) = S(\eta)$. Second, $\bar{S}(\eta)$ can be interpreted as an *update-energy per depth unit* and is the natural quantity that appears in our recursive depth analyses of multi-path networks. Third, as an average, $\bar{S}(\eta)$ is stable under constant-level redefinitions of depth units (e.g., adding or removing $O(1)$ stem/head components), and it yields a single global learning-rate scale that remains width-robust while allowing heterogeneous layers to reallocate update magnitudes. Importantly, AM-$\mu$P does not require every layer to have the same update energy; it controls the typical update energy per depth unit while allowing architecture-dependent redistribution across stems, branches, residual updates, and heads.

A more detailed rationale, including comparisons to alternative aggregations, is deferred to Sec. A.

### 3.3. Depthwise learning-rate scaling for CNNs

**Setup.** We consider 1D/2D CNNs under the conventions in Sec. 3.1. We analyze the one-step update at random initialization under the AM-$\mu$P budget. Recall that $\eta_\star$ denotes the AM-$\mu$P maximal-update learning rate defined in Sec. 3.2, i.e., the (architecture-wide) learning-rate scale that keeps the one-step AM-$\mu$P update budget $\bar{S}$ at $O(1)$ at initialization. Here $B$ denotes the mini-batch size used to form the gradient in this one-step update.

**Boundary fraction for zero padding.** For a spatial index set $\Lambda \subset \mathbb{Z}^d$ and a kernel offset set $\mathcal{K} \subset \mathbb{Z}^d$, define the $\mathcal{K}$-boundary set

$$\partial_{\mathcal{K}}\Lambda \;:=\; \{\, p \in \Lambda : \exists \Delta \in \mathcal{K} \text{ s.t. } p + \Delta \notin \Lambda \,\},$$

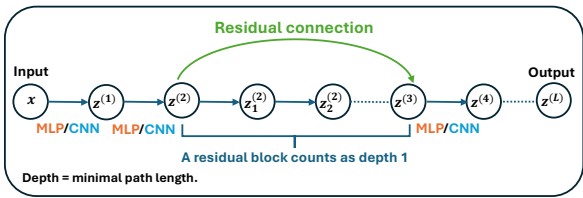

*Figure 2.* Depth convention for residual networks. Depth is defined as the minimal path length. Along the minimal path, each plain layer and each residual block contributes one depth unit. If the backbone has $m$ plain layers and $K$ residual blocks, then the effective depth is $L = m + K$.

and the boundary fraction

$$\mathrm{bdry}(\Lambda, \mathcal{K}) := \frac{|\partial_{\mathcal{K}}\Lambda|}{|\Lambda|}.$$

For circular padding, $\mathrm{bdry}(\Lambda, \mathcal{K}) = 0$.

**Proposition** (Depthwise LR scale for 1D/2D CNNs). *Consider a depth-L CNN with stride $1$, fan-in initialization (Eq. equation 1), and pointwise activation $\sigma$. Assume the homogeneity conditions stated in Sec. 3.1 (up to padding-induced boundary non-uniformity and finite-width effects). Then there exists a constant $\kappa = O(1)$ (depending on $\sigma$ and the initialization statistics such as $q = \mathbb{E}[\sigma'(Z)^2]$, but not on $L$) such that the AM-$\mu$P maximal-update learning rate $\eta_\star$ (Sec. 3.2) satisfies*

$$\eta_\star(L; \{C_\ell, \Lambda_\ell, \mathcal{K}_\ell\}, B) = \kappa\, L^{-3/2}$$
$$\cdot \left(1 + O\left(\max_\ell \frac{1}{C_\ell} + \max_\ell \mathrm{bdry}(\Lambda_\ell, \mathcal{K}_\ell) + \frac{1}{B}\right)\right).$$

*In particular, the leading depth exponent is $-3/2$, and the listed terms only affect the prefactor.*

**Proof.** Deferred to Appendix B.

**Rectangular grids.** We record two common special cases. If $\Lambda_\ell$ is a 1D interval of length $N_\ell$ and $\mathcal{K}_\ell$ has half-span $s_\ell := \max_{\Delta \in \mathcal{K}_\ell} |\Delta|$, then $\mathrm{bdry}(\Lambda_\ell, \mathcal{K}_\ell) = O(s_\ell / N_\ell)$ under zero padding. If $\Lambda_\ell$ is a 2D rectangle of size $H_\ell \times W_\ell$ and $\mathcal{K}_\ell$ has axial half-spans $s_{\ell,h} := \max_{\Delta \in \mathcal{K}_\ell} |\Delta_h|$ and $s_{\ell,w} := \max_{\Delta \in \mathcal{K}_\ell} |\Delta_w|$, then $\mathrm{bdry}(\Lambda_\ell, \mathcal{K}_\ell) = O(s_{\ell,h}/H_\ell + s_{\ell,w}/W_\ell)$.

### 3.4. Depthwise learning-rate scaling for residual networks

We use standard asymptotic notation: $f(L) = \Theta(g(L))$ means that there exist constants $c_1, c_2 > 0$ and $L_0$ such that $c_1 g(L) \le f(L) \le c_2 g(L)$ for all $L \ge L_0$.

**Setup.** We consider a residual-network backbone that consists of $m$ *plain* depth units (e.g., a linear or convolutional layer followed by a pointwise nonlinearity) and $K$ residual blocks inserted along the backbone. A residual block is a depth unit with one residual addition, as defined in Eq. 4.

We measure depth by the minimal path length, counting each plain layer and each residual block as one depth unit (Fig. 2). Accordingly, we define the effective depth as

$$L_{\mathrm{Res}} := m + K.$$

In common ResNet families, the plain part typically includes an input stem and an output head; under our convention these components are simply counted in $m$ rather than being hidden.

For initialization, weights in plain layers use the fan-in rule in Eq. 1. Weights on residual branches use the depth-scaled fan-in initialization in Sec. 3.1, i.e., standard fan-in initialization together with a multiplicative $K^{-1/2}$ scaling on the residual branch. We study the AM-$\mu$P maximal-update learning-rate scale $\eta_\star$ defined in Sec. 3.2, i.e., the learning-rate scale that keeps the one-step AM-$\mu$P update budget $\bar{S}$ at $O(1)$ at initialization.

**Proposition** (Depthwise LR scale for residual networks). *Consider a ResNet-style network of effective depth $L = m + K$ in the above sense. Assume fan-in initialization (Eq. 1) for weights outside residual branches, and the residual-aware depth-scaled fan-in initialization from Sec. 3.1 for residual-branch weights (equivalently, applying standard fan-in initialization and scaling residual-branch weights by $K^{-1/2}$). Then the AM-$\mu$P maximal-update learning rate $\eta_\star$ (Sec. 3.2) satisfies*

$$\eta_\star(L_{\mathrm{Res}}) = \Theta\left(L_{\mathrm{Res}}^{-3/2}\right).$$

*The hidden constants depend on the activation and initialization statistics and on the residual-block template (e.g., the fixed internal structure within each block), but not on the effective depth $L_{\mathrm{Res}}$.*

**Proof.** Deferred to Appendix C.

### 3.5. Depthwise learning-rate scaling for Transformers

**Setup.** We consider Transformer architectures under the conventions in Sec. 3.1 and analyze the one-step update at random initialization under the AM-$\mu$P budget. We denote the resulting effective depth by $L_{\mathrm{tr}}$.

**Proposition** (Depthwise LR scale for Transformers). *Consider a Transformer of effective depth $L_{\mathrm{tr}}$, with fan-in initialization (Eq. 1) for attention and FFN linear maps, following Sec. 3.1. Under the mean-field, weak-dependence, and LayerNorm-stabilized tangent-propagation assumptions used in our AM-$\mu$P analysis, there exists a constant $\kappa$ independent of $L_{\mathrm{tr}}$ such that the AM-$\mu$P maximal-update learning rate $\eta_\star$ (Sec. 3.2) satisfies*

$$\eta_\star(L_{\mathrm{tr}}) = \Theta\left(L_{\mathrm{tr}}^{-3/2}\right).$$

*The hidden constants depend on the activation and initialization statistics and on the fixed Transformer-block template, but not on $L_{\mathrm{tr}}$.*

*Proof.* Deferred to Appendix D.

# 4. Experiments

## 4.1. General Protocol

To evaluate the predicted depth dependence of the base learning rate, we design CNNs, ResNets, and ViTs with varying effective depths and measure the learning rate selected by an early-training grid search. Inspired by Chen et al. (2024), we conduct experiments on three image classification datasets: CIFAR-10, CIFAR-100 (Krizhevsky, 2009), and a subset of ImageNet (Deng et al., 2009), which span a range of task difficulties, allowing us to demonstrate the robustness of our theory across different data regimes.

For each network depth $L$, we perform a logarithmic grid search over learning rates $\eta$ and record $\eta^\star$, the one-epoch optimum that minimizes the training loss. We treat this grid-searched one-epoch optimum as an empirical proxy for the maximal-update learning-rate scale $\eta^\star$ defined in Sec. 3.2.[3] Thus the sweep is used to calibrate the base learning-rate scale, not to claim optimality of a complete learning-rate schedule. We then fit the depth-dependent scaling law on the log–log scale via

$$\log_{10} \eta^\star = \beta_0 + \alpha \log_{10} L + \varepsilon,$$

and report the fitted slope $\hat{\alpha}$. All experiments use standard multi-class cross-entropy loss with mean reduction.[4] To ensure statistical robustness, we repeat each experiment with three random seeds, compute the depth-wise mean $\pm$ 95% confidence interval for $\log_{10} \eta^\star$, and perform weighted least squares fitting with weights inversely proportional to the sample variance at each depth.

Table 1 summarizes the fitted depth exponents $\hat{\alpha}$ for baseline configurations. The leading-order theoretical prediction is $\alpha = -1.5$. Across the 9 architecture–dataset combinations, the empirical exponents range from $-1.18$ to $-1.57$, with a mean of $-1.38$. Although the experiments are finite-depth while the theory is asymptotic, the independently tuned optima consistently organize around a common power law across CNNs, ResNets, and ViTs. We view this as strong evidence for the shared leading-order $L^{-3/2}$ depth scale, with the remaining variation reflecting finite-depth offsets, fixed stem/head components, and architecture-dependent constants. Additional transfer checks in Appendix E.4 support the same depth signal across ViT proxy widths and later-epoch LR selection. In

a direct zero-shot test, calibrating a source LR at one ViT depth and transferring it with the $L^{-3/2}$ rule reduces the median log-LR error to independently tuned oracle LRs from 0.314 to 0.057 decades and gives lower unrounded epoch-3 transfer loss than raw transfer on $6/7$ non-source target depths.

*Table 1.* **Main results: fitted depth exponents $\hat{\alpha}$ for baseline configurations.** All models use SGD optimizer without momentum. CNN and ResNet use ReLU activation without normalization or dropout. ViT uses Post-LN configuration. The theoretical prediction is $\alpha = -1.5$.

| Model | CIFAR-10 | CIFAR-100 | ImageNet |
|---|---|---|---|
| CNN | $-1.339$ | $-1.392$ | $-1.329$ |
| ResNet | $-1.435$ | $-1.355$ | $-1.567$ |
| ViT | $-1.441$ | $-1.371$ | $-1.178$ |
| *Theory* | | $\alpha = -1.5$ | |

## 4.2. Convolutional Networks

We first validate our theory on plain CNNs(O'Shea & Nash, 2015; Krizhevsky et al., 2012). Following the homogeneous CNN conventions in Sec. 3.1, we use $L$ identical 2D `conv`+$\sigma$ blocks with stride 1 and circular padding, followed by global average pooling and a linear classifier. Networks are initialized with He fan-in and trained using SGD without momentum (batch size 128). We sweep $\eta$ over 80 log-spaced points from $10^{-4}$ to $10^1$ and take $\eta^\star$ to be the one-epoch training-loss minimizer.

As shown in Fig. 3(a), the optimal learning rate $\eta^\star$ exhibits a clear power-law relationship with depth on CIFAR-10, yielding a fitted slope of $\hat{\alpha} = -1.339$. Notably, the exponents remain stable across datasets: $-1.392$ on CIFAR-100 and $-1.329$ on ImageNet (Table 1), confirming the cross-dataset robustness of our theory.

## 4.3. Residual Networks

We define depth $L$ by the number of residual blocks, each containing a $3 \times 3$ conv layer (64 channels, stride 1) with an identity skip. Networks are initialized with scaled He fan-in, where conv weights on residual branches are multiplied by $1/\sqrt{K}$ for $K$ blocks to stabilize variance.

As shown in Fig. 3(b), the optimal learning rate follows a clear power law with $\hat{\alpha} = -1.435$ on CIFAR-10, closely matching our theoretical prediction of $-1.5$ and confirming that residual connections do not alter the depth exponent. In contrast, PathSum (Chen et al., 2024) shows increasing deviation from empirical optima at larger depths. Across datasets, the exponents remain consistent: $-1.355$ on CIFAR-100 and $-1.567$ on ImageNet.

---

[3]We adopt a single-epoch protocol for computational efficiency and comparability, consistent with the architecture-aware scaling protocol (Chen et al., 2024) and the $\mu P$ perspective that optimal learning rates are primarily governed by early-training dynamics (Jelassi et al., 2023).

[4]See Appendix G for compatibility between CE-based experiments and our MSE-based derivation.

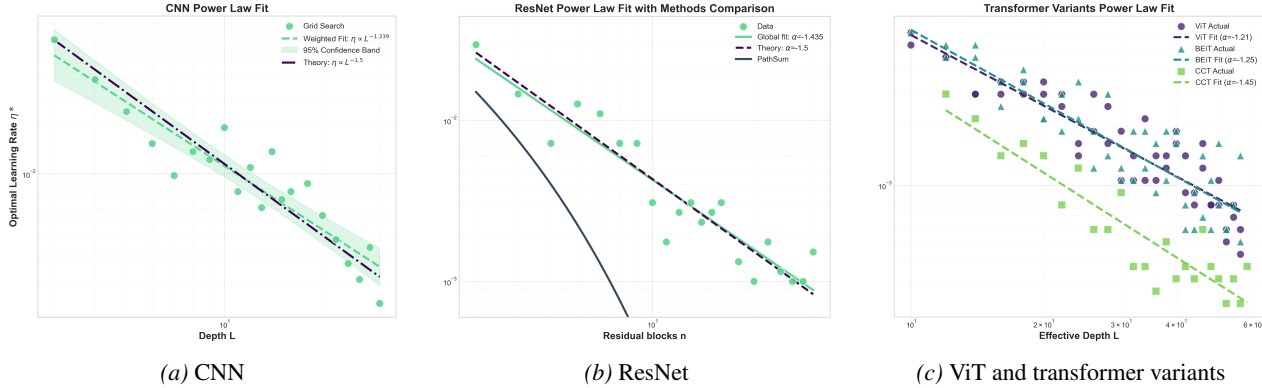

|     |     |     |
| :-: | :-: | :-: |
| *(a)* CNN | *(b)* ResNet | *(c)* ViT and transformer variants |

*Figure 3.* **Depth–LR scaling on CIFAR-10.** (a) CNN: grid-searched optima with 95% CIs and weighted global fit ($\hat{\alpha} = -1.339$). (b) ResNet: our AM-$\mu$P theory ($\hat{\alpha} = -1.435$) closely matches empirical data, while PathSum (Chen et al., 2024) shows increasing deviation at larger depths. (C) Transformer: Depth–LR scaling for ViT variants ViT ($\hat{\alpha} = -1.44$), BEiT ($\hat{\alpha} = -1.35$), and CCT ($\hat{\alpha} = -1.45$) all exhibit clear power-law scaling consistent with our theoretical prediction of $-1.5$.

### 4.4. Vision Transformers

Finally, we extend our analysis to Vision Transformers. For vanilla ViT(Dosovitskiy, 2020), we adopt the standard architecture with patch embedding, learnable positional encoding, and a classification token. Following the Transformer depth convention in Sec. 3.5, we measure depth by the effective Transformer depth $L_{\text{tr}}$. For CIFAR datasets, we use patch size 4, embedding dimension 384, and 6 attention heads; for ImageNet, we use patch size 16, embedding dimension 768, and 12 attention heads. The learning rate is searched over 80 log-spaced points from $10^{-5}$ to $10^0$.

The baseline ViT (Post-LN) achieves $\hat{\alpha} = -1.441$ on CIFAR-10, $-1.371$ on CIFAR-100, and $-1.178$ on ImageNet (Table 1).

**ViT variants.** Beyond vanilla ViT, we evaluate two representative variants: BEiT (Bao et al., 2022) and CCT (Hassani et al., 2021). BEiT replaces absolute positional embeddings with relative position bias in the attention mechanism. CCT replaces the patch embedding with a convolutional tokenizer and uses sequence pooling instead of the class token, thereby combining the inductive biases of CNNs with the global modeling capacity of transformers.

As shown in Fig. 3 (c), all three variants exhibit clear power-law scaling with fitted exponents of $\hat{\alpha} = -1.44$ (ViT), $-1.35$ (BEiT), and $-1.45$ (CCT). Notably, CCT achieves the closest agreement with our theoretical prediction of $-1.5$, with only 3.3% deviation.

The loss landscape analysis in Fig. 4 corroborates these findings. For all three variants, the loss curves exhibit systematic stratification: as depth increases (from purple to yellow), the loss-minimizing learning rate shifts leftward, and the curves become steeper in the high learning rate regime. This consistent leftward shift across Transformer variants supports the shared leading-order depth trend predicted by the theory.

We attribute CCT's strong agreement with theory to its hybrid architecture. CCT's convolutional tokenizer introduces CNN-like components at the input stage, while its transformer encoder captures global dependencies. Since both CNNs (Section 4.2) and transformers independently follow the $\eta^{\star} \propto L^{-3/2}$ scaling law, it is natural that their combination in CCT also adheres to this relationship. This observation suggests that hybrid architectures combining analyzed building blocks are a natural target for future tests of the same effective-depth rule.

### 4.5. Ablation Studies

To assess the robustness of our theoretical predictions, we conduct systematic ablation studies examining the effects of optimizers, activation functions, normalization, and regularization. Table 3 in Appendix E.1 presents 9 representative configurations spanning CNNs, ResNets, and ViTs.

**Overall robustness.** All tested configurations yield exponents within the range $[-1.8, -1.1]$, with a mean of $-1.39$, demonstrating strong agreement with the theoretical prediction of $\alpha = -1.5$. This consistency across diverse architectural and optimization choices validates the broad applicability of our depth scaling law.

**Activation functions.** Comparing ReLU and GELU on CNN/CIFAR-10, we observe $\hat{\alpha} = -1.339$ versus $-1.379$ (3% difference), showing the choice of activation has minimal impact. This aligns with our theory, where activations enter primarily through the gating factor $q = \mathbb{E}[\sigma'(Z)^2]$.

**Optimizers.** Switching from SGD to Adam reduces $|\hat{\alpha}|$ by 10–12% on both CNNs ($-1.339 \rightarrow -1.207$) and ResNets ($-1.435 \rightarrow -1.269$). While Adam's adaptive learning rates partially compensate for depth scaling, the power-law relationship persists, confirming that depth-dependent initialization remains essential even with adaptive optimizers.

**Normalization and regularization.** BatchNorm on

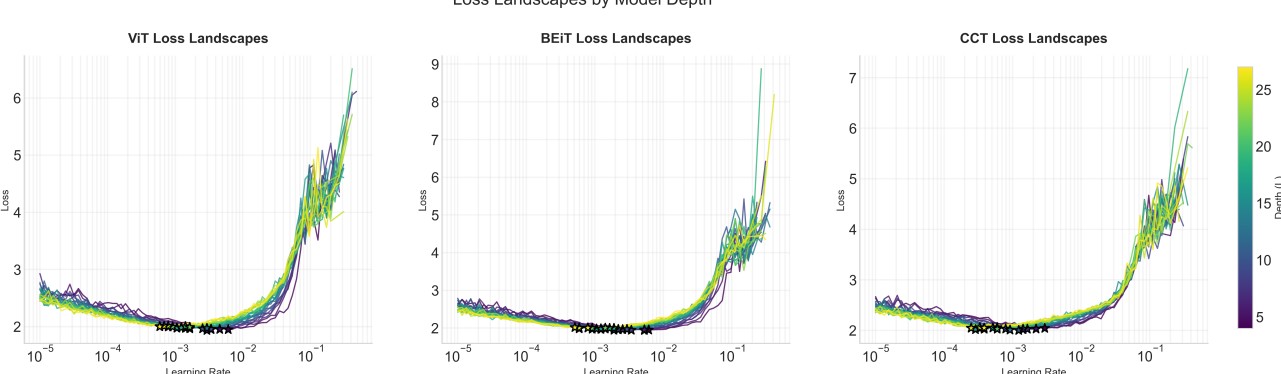

*Figure 4.* **Loss landscapes for ViT variants on CIFAR-10.** Color encodes depth (purple: shallow, yellow: deep). All three variants show systematic leftward shifts in optimal LR as depth increases, consistent with the predicted depth–LR scaling relationship.

ResNets increases $|\hat{\alpha}|$ from $-1.435$ to $-1.701$, while dropout increases it to $-1.568$, both bringing the exponent closer to the theoretical $-1.5$. In ViTs, Pre-LN and Post-LN yield similar exponents ($-1.131$ vs $-1.178$, 4.0% difference), indicating that normalization placement has minimal effect on depth scaling.

These ablations confirm that the $L^{-3/2}$ scaling law provides a robust first-order approximation across diverse training configurations, with secondary modulations typically within $\pm 20\%$.

## 5. Conclusion

We studied depthwise learning-rate scaling for dense non-recurrent multi-path neural networks. Using an architecture-dependent effective-depth convention and a network-wide update-energy criterion, we derived a shared leading-order -3/2 law for the base learning-rate scale in CNNs, ResNets, and Transformers. This gives a simple operational rule: calibrate the learning rate at a reference effective depth $L_0$ and rescale it to a new effective depth by

$$\eta(L) = \eta(L_0) \left( \frac{L}{L_0} \right)^{-3/2}.$$

Empirically, independently tuned learning-rate optima organize around this law across architectures and datasets. Additional checks across proxy widths, later epochs, direct zero-shot transfer, and common training variants support the rule as a practical first-order learning-rate rescaling method. The finite-depth variation we observe is consistent with fixed stems and heads, block-template constants, optimizer effects, and other architecture-dependent prefactors.

## 6. Limitations and Future Work

**Training dynamics and optimizers.** Our analysis identifies a base learning-rate scale from one-step representa-

tion updates at stabilizing random initialization. This focus is natural for maximal-update scaling, since early activation, gradient, and normalization statistics are largely controlled by architecture and initialization. At finite depth, tuned optima may fluctuate around the leading law because fixed stems, heads, block-template constants, and data-dependent effects are not asymptotically averaged out. Later in training, these quantities become increasingly data-dependent, coupling the appropriate learning rate to the training state and schedule; a natural next step is to combine the present depthwise scale with warmup, cosine decay, adaptive target learning rates, or feedback-based control views over the full training trajectory. In practice, calibration constants should be transferred within the same optimizer and schedule family. Extending the framework to more optimizers, including Adam-style preconditioning and Muon-style matrix updates, is a promising direction.

**Beyond fixed architecture families.** The formal derivations in this paper cover CNNs, ResNets, and Transformers under the effective-depth conventions studied here. A longer-term goal is to obtain a more general graph-level scaling rule that applies across heterogeneous computation graphs. Architectures with concatenative skips, multi-scale fusion, graph-transformer hybrids, neural fields, Mixture-of-Experts, or learned aggregation may require merge-aware or graph-aware depth coefficients beyond the current minimal-path convention. Testing and refining the rule on more realistic tasks and models, including language, audio, multimodal learning, reinforcement learning, and pre-trained or fine-tuned systems, is also an important future direction.

**Broader impact.** The main expected benefit of this work is to reduce repeated learning-rate sweeps when scaling models in depth, lowering tuning cost, engineering effort, and associated energy use. Because cheaper scaling methods can also lower the barrier to training large models, responsible deployment of downstream systems remains important.

## Impact Statement

This paper studies learning-rate scaling rules for training deeper neural networks. The main positive impact is to reduce the cost of repeated learning-rate sweeps when scaling models in depth, which can lower compute usage, engineering effort, and associated energy consumption. By making depth-scaling experiments less dependent on expensive trial-and-error tuning, the proposed rule may also make such experiments more accessible to resource-constrained research groups.

The work is methodological and does not introduce new datasets, deploy models, or target a specific application domain. Its risks are therefore mostly indirect. More efficient tuning methods can lower the barrier to training larger models, including models that may be misused in downstream applications. We therefore view this work as complementary to standard responsible deployment practices for the models trained using these methods.

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

# Notation

*Table 2.* Notation summary.

| Symbol | Meaning |
| --- | --- |
| **Depth / architecture** | |
| $L$ | Network effective depth (number of depth units). |
| $\ell$ | Depth-unit index, $\ell = 1, \ldots, L$. |
| $m$ | Number of plain (non-residual) depth units along the minimal path. |
| $K$ | Number of residual blocks along the minimal path. |
| $L_{\text{res}}$ | ResNet effective depth, $L_{\text{res}} := m + K$. |
| $L_{\text{tr}}$ | Transformer effective depth (2 depth units per transformer block). |
| **Data / optimization** | |
| $x$ | Input example. |
| $y$ | Target / label. |
| $B$ | Mini-batch (set); when used as a scalar, it denotes mini-batch size. |
| $\mathcal{L}_B(\theta)$ | Training loss on mini-batch $B$. |
| $\theta$ | All trainable parameters; $\theta^+$ parameters after one (S)GD step. |
| $\eta$ | Base learning rate. |
| $\eta^\star$ | maximal-update learning rate defined by $\bar{S}(\eta^\star) = 1$. |
| **Forward pass / initialization** | |
| $z^{(\ell)}(x;\theta)$ | Pre-activations at depth unit $\ell$ for input $x$. |
| $z_i^{(\ell)}(x;\theta)$ | Scalar coordinate of $z^{(\ell)}$ (unit index $i$). |
| $\sigma(\cdot)$ | Pointwise activation; $\sigma'(\cdot)$ its derivative. |
| $q$ | Gating factor $q := \mathbb{E}[\sigma'(Z)^2]$ for $Z \sim \mathcal{N}(0, 1)$ (for ReLU, $q = 1/2$). |
| $\text{fan}_{\text{in}}(W)$ | Fan-in of a weight tensor. |
| $W^{(\ell)}$ | Weight matrix / kernel tensor at depth unit $\ell$. |
| $W_{\text{res}}$ | Residual-branch weights . |
| **One-step updates / AM-$\mu$P** | |
| $\partial_a$ | Partial derivative $\partial/\partial\theta_a$ for scalar parameter $\theta_a$. |
| $\Delta\theta_a$ | One-step parameter change, $\Delta\theta_a := \theta_a^+ - \theta_a$. |
| $\Delta z_i^{(\ell)}(x)$ | One-step change of pre-activation coordinate $i$ at depth $\ell$. |
| $S_\ell(\eta)$ | Layerwise second moment $S_\ell(\eta) := \mathbb{E}[(\Delta_B z_i^{(\ell)}(x))^2]$. |
| $\bar{S}(\eta)$ | Network-wide average second moment, $\bar{S}(\eta) := \frac{1}{L}\sum_{\ell=1}^{L} S_\ell(\eta)$. |
| **CNN-specific** | |
| $d$ | Spatial dimension ($d = 1$ or 2). |
| $C_\ell$ | Number of channels at depth $\ell$. |
| $\Lambda_\ell$ | Spatial index set at depth $\ell$; $N_\ell := |\Lambda_\ell|$. |
| $N_\ell$ | Number of spatial locations at depth $\ell$ (e.g., $N_\ell = H_\ell W_\ell$ in 2D). |
| $H_\ell, W_\ell$ | Height/width of a 2D feature map at depth $\ell$. |
| $K_\ell$ | Kernel offset set (subset of $\mathbb{Z}^d$); $k_\ell := |K_\ell|$. |
| $k_\ell$ | Kernel size / number of offsets ($k_\ell = |K_\ell|$). |
| $p$ | Spatial location index ($p \in \Lambda_\ell$). |
| $\Delta$ | Kernel offset ($\Delta \in K_\ell$). |
| $W_{j,i,\Delta}^{(\ell)}$ | Conv weight from input channel $i$ to output channel $j$ at offset $\Delta$. |
| $M_\ell$ | Effective width of a conv layer, $M_\ell := C_\ell N_\ell$. |
| $\partial_K \Lambda$ | $K$-boundary set under zero padding: $\{p \in \Lambda : \exists \Delta \in K, \ p + \Delta \notin \Lambda\}$. |
| $\text{bdry}(\Lambda, K)$ | Boundary fraction $\text{bdry}(\Lambda, K) := |\partial_K \Lambda|/|\Lambda|$. |
| $s_\ell$ | 1D kernel half-span $s_\ell := \max_{\Delta \in K_\ell} |\Delta|$. |
| $s_{\ell,h}, s_{\ell,w}$ | 2D axial half-spans $\max|\Delta_h|$ and $\max|\Delta_w|$. |
| **ResNets / Transformers (selected)** | |
| $F_\ell(\cdot)$ | Residual branch mapping in depth unit $\ell$. |
| $\text{Attn}, \text{FFN}$ | Attention branch and feedforward branch in a Transformer block. |
| $\text{LN}$ | LayerNorm; $\gamma, \beta$ are its affine parameters; $\epsilon$ is the numerical constant. |
| $u^{(\ell)}$ | Pre-LN residual sum in post-norm analysis: $u^{(\ell)} = z^{(\ell-1)} + F_\ell(z^{(\ell-1)})$. |
| $T_\ell(\mu_1, \mu_2)$ | Averaged overlap of directional derivatives at depth $\ell$ (used in proofs). |
| **Asymptotic notation** | |
| $O(\cdot)$ | Big-O: $f(s) = O(g(s))$ if $\exists c > 0, s_0$ such that $|f(s)| \le c|g(s)|$ for all $s \ge s_0$. |
| $\Theta(\cdot)$ | Big-Theta: $f(s) = \Theta(g(s))$ if $\exists c_1, c_2 > 0, s_0$ such that $c_1 g(s) \le f(s) \le c_2 g(s)$ for all $s \ge s_0$. |
| $\propto$ | Asymptotic proportionality: $f \propto g$ means $f(s)/g(s) \to \kappa \in (0, \infty)$. |

# A. Rationale for the Arithmetic Mean in $\mu$P

This appendix supplements Sec. 3.2 and gives a more formal rationale for using an *arithmetic mean* to aggregate layerwise maximal-update energies in AM-$\mu$P.

**Setup and notation.** We follow Sec. 3.2. Let $\Delta z_i^{(\ell)}(x)$ denote the one-step change of a representative preactivation coordinate $z_i^{(\ell)}(x)$ at depth unit $\ell$ after one gradient step at initialization (with $x \sim \mathcal{D}$). Define the layerwise energy

$$S_\ell \coloneqq \mathbb{E}\Big[\big(\Delta z_i^{(\ell)}(x)\big)^2\Big], \qquad \bar{S} \coloneqq \frac{1}{L}\sum_{\ell=1}^{L} S_\ell, \quad (10)$$

where the expectation is taken over data (and, when relevant, initialization and mini-batch sampling). AM-$\mu$P fixes a network-level budget $\bar{S} = C$ for an $O(1)$ constant $C$ (and we may set $C = 1$ by absorbing constants into the learning-rate scale).

**(A1) Merge consistency and characterization.** Let $\mathcal{M}(S_1, \ldots, S_L)$ be a network-level aggregator that summarizes the collection $\{S_\ell\}_{\ell=1}^{L}$ into a single scale used to set the global learning rate. We require three minimal properties:

- **Permutation invariance.** $\mathcal{M}$ is invariant under permutations of the inputs.

- **Scale equivariance.** For any $c > 0$, $\mathcal{M}(cS_1, \ldots, cS_L) = c\,\mathcal{M}(S_1, \ldots, S_L)$.

- **Merge consistency.** For any partition $\{1, \ldots, L\} = \bigsqcup_{j=1}^{k} G_j$, define each group's *per-layer* energy by

$$m_j \coloneqq \frac{1}{|G_j|}\sum_{\ell \in G_j} S_\ell,$$

so that the total energy of group $G_j$ is preserved when replacing it by $|G_j|$ identical copies of $m_j$. Merge consistency requires

$$\mathcal{M}(S_1, \ldots, S_L) = \mathcal{M}(\underbrace{m_1, \ldots, m_1}_{|G_1|}, \ldots, \underbrace{m_k, \ldots, m_k}_{|G_k|}).$$
$$(11)$$

In addition, we adopt the standard normalization for a "mean":
$$\mathcal{M}(c, \ldots, c) = c, \qquad \forall c \geq 0. \quad (12)$$

*Consequence.* Applying equation 11 with the single group $G_1 = \{1, \ldots, L\}$ yields $\mathcal{M}(S_1, \ldots, S_L) = \mathcal{M}(\bar{S}, \ldots, \bar{S})$. By equation 12, this equals $\bar{S}$. Hence the arithmetic mean is the unique aggregator satisfying the above properties.

**(A2) Why $\bar{S}$ is the right "energy budget."** The quantity $\bar{S}$ has a direct probabilistic interpretation: if $\ell$ is drawn uniformly from $\{1, \ldots, L\}$, then

$$\bar{S} = \mathbb{E}_\ell\, S_\ell = \mathbb{E}_{\ell,x}\Big[\big(\Delta z_i^{(\ell)}(x)\big)^2\Big].$$

Thus fixing $\bar{S} = O(1)$ controls the *typical* depth-unit update energy, while allowing heterogeneous allocation across layers, which is essential in multi-path architectures (residual accumulation, convolutional coupling, attention/FFN branches).

**(A3) Heterogeneity: what is controlled and what is not.** The arithmetic mean is monotone and satisfies the sharp bounds
$$\min_\ell S_\ell \leq \bar{S} \leq \max_\ell S_\ell.$$

Therefore, when layerwise energies remain within constant factors, $\bar{S}$ remains within the same range. More importantly, if some layers become anomalously large, $\bar{S}$ *reflects* this increase, which is appropriate when $\bar{S}$ is meant to represent an honest second-moment budget. This contrasts with multiplicative aggregates that can hide large outliers.

**(A4) Failure of the geometric mean: multiplicative cancellation.** Let $G \coloneqq \left(\prod_{\ell=1}^{L} S_\ell\right)^{1/L}$. Consider $S = (\varepsilon, \varepsilon^{-1}, 1, \ldots, 1)$ with $\varepsilon \downarrow 0$. Then $G = 1$ stays constant, while
$$\bar{S} = \frac{1}{L}\big(\varepsilon + \varepsilon^{-1} + L - 2\big) \to \infty.$$

Hence fixing $G$ does not control the additive second-moment budget $\sum_\ell S_\ell$. In heterogeneous settings, $G$ can severely underestimate the presence of large-update layers.

**(A5) Failure of the harmonic mean: hypersensitivity to small layers.** Let $H \coloneqq \left(\frac{1}{L}\sum_{\ell=1}^{L} S_\ell^{-1}\right)^{-1}$. A direct differentiation gives

$$\frac{\partial H}{\partial S_i} = \frac{H^2}{L}\cdot\frac{1}{S_i^2} > 0, \qquad S_i \downarrow 0 \Rightarrow \frac{\partial H}{\partial S_i} \uparrow \infty.$$

Thus the harmonic mean becomes dominated by the smallest layers and can force disproportionate allocation decisions when used as a global budget.

**(A6) Granularity and effective depth (split/merge compatibility).** In our paper, "depth units" are defined at an architecture-dependent granularity (Sec. 3.1), e.g., residual additions along the minimal path. One may equivalently analyze the same network at a coarser granularity by grouping depth units into blocks (e.g., treating a Transformer block as two sequential residual updates). Merge consistency equation 11 formalizes the requirement that

the global budget should be invariant to such regrouping as long as group totals are preserved via per-unit averages. The arithmetic mean satisfies this invariance exactly, whereas geometric/harmonic means generally do not.

**(A7) Homogeneous limit.** When $S_\ell \approx S$ for all $\ell$, the constraint $\bar{S} = C$ is equivalent to $S_\ell \approx C$, recovering the classical per-layer maximal-update viewpoint in settings where it is well posed.

*Takeaway.* The arithmetic mean is the only network-level aggregator that is simultaneously compatible with additive second-moment budgeting and consistent under split/merge (coarse-graining) of depth units. This motivates the AM-$\mu$P constraint $\bar{S} = C$ used throughout the paper.

## B. Proof of Proposition 3.3 for CNNs

**Setup and scope.** We prove the depth exponent in Proposition 3.3 by analyzing the *first (linearized) SGD step at initialization*, which is the standard setting for maximal-update calculations. We work under the CNN conventions in Sec. 3.1: stride 1, pointwise nonlinearity $\sigma$, fan-in initialization (Eq. equation 1), and the homogeneous (stationary) setting under circular padding. The 2D case follows by the same counting argument and is stated at the end.

For a layer $\ell$, let $C_\ell$ be the channel count, $\Lambda_\ell$ the spatial index set, $N_\ell := |\Lambda_\ell|$, and define the effective width

$$M_\ell := C_\ell N_\ell.$$

We flatten a channel–position pair into a single unit index $a \in \{1, \ldots, M_\ell\}$ when convenient; this is purely notational.

**One-step pre-activation increments and the AM-$\mu$P budget.** Let $\theta$ collect all trainable parameters (weights in convolutional layers and the linear head). Write $\theta^+ = \theta + \Delta\theta$ for the parameters after one SGD step with learning rates $\{\eta_\mu\}$ at each scalar parameter direction $\mu$. For each depth unit $\ell$ and input $x$, define the (linearized) pre-activation increment

$$\Delta z^{(\ell)}(x) := \sum_{\mu \in \mathcal{P}_{\leq \ell}} \partial_\mu z^{(\ell)}(x) \, \Delta\mu,$$

where $\mathcal{P}_{\leq \ell}$ denotes parameters that influence $z^{(\ell)}$ (i.e., parameters in layers up to $\ell$), and $\partial_\mu$ denotes the derivative w.r.t. the scalar parameter $\mu$.

We use the layerwise energy (averaged over units)

$$S_\ell := \mathbb{E}\left[\frac{1}{M_\ell} \sum_{a=1}^{M_\ell} \left(\Delta z_a^{(\ell)}\right)^2\right], \qquad \bar{S} := \frac{1}{L} \sum_{\ell=1}^{L} S_\ell.$$

The AM-$\mu$P maximal-update learning-rate scale $\eta_\star$ is the scale of $\{\eta_\mu\}$ that keeps $\bar{S} = \Theta(1)$ at initialization (Sec. 3.2).

**Loss model (for analytic convenience).** As in prior one-step maximal-update analyses, we use a mean-squared error (MSE) surrogate to make the second-moment expansion explicit. This choice only affects constant prefactors and does not change the depth exponent. (If desired, a short cross-entropy-at-initialization justification can be included elsewhere.)

Let the output layer have dimension $M_{L+1}$ and define the per-sample loss

$$\mathcal{L} = \frac{1}{2M_{L+1}} \left\| z^{(L+1)}(x) - y \right\|_2^2.$$

We assume $\mathbb{E}[y] = 0$ and $\mathrm{Var}(y_t) = \sigma_y^2$ for each output coordinate $t$, independent across $t$ and independent of the network at initialization.

### B.1 A top-layer reduction identity for CNN Jacobian overlaps

For two parameter directions $\mu_1, \mu_2$ and a depth $h$, define the averaged Jacobian overlap

$$T_h(\mu_1, \mu_2) := \frac{1}{M_h} \sum_{a=1}^{M_h} \partial_{\mu_1} z_a^{(h)} \, \partial_{\mu_2} z_a^{(h)}. \qquad (13)$$

**Lemma 1** (Top-layer reduction for CNN overlaps). *Assume ReLU ($q = \mathbb{E}[\sigma'(Z)^2] = 1/2$), stride 1, circular padding, and independent, zero-mean fan-in initialization (Eq. equation 1). Fix integers $0 \leq \ell < h \leq L+1$. For any parameter directions $\mu_1, \mu_2$ that belong to layers at most $\ell$,*

$$\mathbb{E}\left[T_h(\mu_1, \mu_2) \,\middle|\, z^{(\ell)}\right] = T_\ell(\mu_1, \mu_2). \qquad (14)$$

*Proof.* We follow the same induction structure as the sequential-network argument and only highlight CNN-specific points.

*Step 1: one-layer expansion.* Condition on $z^{(h-1)}$ and take expectation over the weights of layer $h$ only. For $\mu \leq \ell < h$, the chain rule gives

$$\partial_\mu z^{(h)} = W^{(h)} \left(\sigma'(z^{(h-1)}) \odot \partial_\mu z^{(h-1)}\right),$$

where $W^{(h)}$ is the convolution operator (flattened as a sparse matrix), and $\odot$ denotes pointwise multiplication.

By independence and zero mean of distinct kernel parame-

ters, only diagonal terms survive, yielding

$$\mathbb{E}\Big[T_h(\mu_1, \mu_2) \,\Big|\, z^{(h-1)}\Big]$$
$$= \frac{1}{q} \cdot \frac{1}{M_{h-1}} \sum_{a=1}^{M_{h-1}} \sigma'\big(z_a^{(h-1)}\big)^2 \,\partial_{\mu_1} z_a^{(h-1)} \,\partial_{\mu_2} z_a^{(h-1)}. \tag{15}$$

Here the factor $1/q$ comes from the fan-in variance in Eq. equation 1, and the kernel-size dependence cancels because under circular padding each spatial location is visited exactly $k_h$ times, matching the fan$_{\text{in}}$ factor.

*Step 2: averaging out the gate using symmetry.* Now condition on the earlier representation $z^{(\ell)}$ with $\ell < h-1$. Under ReLU and symmetric (zero-mean) initialization, the distribution of the pre-activations $z^{(h-1)}$ given $z^{(\ell)}$ is symmetric about 0. Moreover, flipping the sign of all weights in layer $(h-1)$ flips $z^{(h-1)}$, while simultaneously flipping each $\partial_\mu z^{(h-1)}$ for $\mu \leq \ell$. Hence the product $\partial_{\mu_1} z_a^{(h-1)} \partial_{\mu_2} z_a^{(h-1)}$ is *even* under the sign flip, while $\sigma'(z_a^{(h-1)})^2 = \mathbf{1}\{z_a^{(h-1)} > 0\}$ is *odd* in the sense that it swaps with $\mathbf{1}\{z_a^{(h-1)} < 0\}$. This implies the factorization

$$\mathbb{E}\Big[\sigma'(z_a^{(h-1)})^2 \,\partial_{\mu_1} z_a^{(h-1)} \,\partial_{\mu_2} z_a^{(h-1)} \,\Big|\, z^{(\ell)}\Big]$$
$$= q \cdot \mathbb{E}\Big[\partial_{\mu_1} z_a^{(h-1)} \,\partial_{\mu_2} z_a^{(h-1)} \,\Big|\, z^{(\ell)}\Big].$$

Plugging this into Eq. equation 15 yields

$$\mathbb{E}\Big[T_h(\mu_1, \mu_2) \,\Big|\, z^{(\ell)}\Big] = \mathbb{E}\Big[T_{h-1}(\mu_1, \mu_2) \,\Big|\, z^{(\ell)}\Big].$$

*Step 3: induction from $h$ down to $\ell$.* Iterating the last identity from $h$ down to $\ell + 1$ proves $\mathbb{E}[T_h(\mu_1, \mu_2) \mid z^{(\ell)}] = T_\ell(\mu_1, \mu_2)$, as claimed. $\square$

**Remark** (Zero padding and boundary corrections). *If circular padding is replaced by zero padding, the uniform coverage argument is violated only near the spatial boundary. Using the boundary fraction* $\mathrm{bdry}(\Lambda_h, \mathcal{K}_h)$ *defined in Sec. 3.3, Eq. equation 15 acquires an additive error term of order* $O(\mathrm{bdry}(\Lambda_h, \mathcal{K}_h))$. *This produces only lower-order multiplicative corrections to* $\eta_\star$ *and does not change the depth exponent.*

## B.2 A/B decomposition for one-step pre-activation changes

Define the output-layer quantities

$$T_{L+1}(\mu_1, \mu_2) := \frac{1}{M_{L+1}} \sum_{t=1}^{M_{L+1}} \partial_{\mu_1} z_t^{(L+1)} \,\partial_{\mu_2} z_t^{(L+1)}, \tag{16}$$

$$S_{L+1}(\mu_1, \mu_2) := \frac{1}{M_{L+1}^2} \sum_{t_1=1}^{M_{L+1}} \sum_{t_2=1}^{M_{L+1}}$$
$$\Big(\partial_{\mu_1} z_{t_1}^{(L+1)} \, z_{t_1}^{(L+1)}\Big)\Big(\partial_{\mu_2} z_{t_2}^{(L+1)} \, z_{t_2}^{(L+1)}\Big). \tag{17}$$

**Lemma 2** (Second-moment decomposition). *Fix a depth $\ell \leq L$ and a unit $a$ in layer $\ell$. Under the MSE model above, after one SGD step at initialization,*

$$\mathbb{E}\Big[\big(\Delta z_a^{(\ell)}\big)^2\Big] = A_{\text{cnn}}^{(\ell)} + B_{\text{cnn}}^{(\ell)}, \tag{18}$$

*where*

$$B_{\text{cnn}}^{(\ell)} := \sigma_y^2 \,\mathbb{E}\Bigg[ \sum_{\mu_1, \mu_2 \in \mathcal{P}_{\leq \ell}}$$
$$\eta_{\mu_1} \eta_{\mu_2} \,\partial_{\mu_1} z_a^{(\ell)} \,\partial_{\mu_2} z_a^{(\ell)} \, T_{L+1}(\mu_1, \mu_2)\Bigg], \tag{19}$$

$$A_{\text{cnn}}^{(\ell)} := \mathbb{E}\Bigg[ \sum_{\mu_1, \mu_2 \in \mathcal{P}_{\leq \ell}}$$
$$\eta_{\mu_1} \eta_{\mu_2} \,\partial_{\mu_1} z_a^{(\ell)} \,\partial_{\mu_2} z_a^{(\ell)} \, S_{L+1}(\mu_1, \mu_2)\Bigg]. \tag{20}$$

*Proof.* Let $g := z^{(L+1)}(x) - y$. The gradient of the MSE loss satisfies

$$\partial_\mu \mathcal{L} = \frac{1}{M_{L+1}} \sum_{t=1}^{M_{L+1}} g_t \,\partial_\mu z_t^{(L+1)}, \qquad \Delta\mu = -\eta_\mu \,\partial_\mu \mathcal{L}.$$

Substitute $\Delta\mu$ into $\Delta z_a^{(\ell)} = \sum_{\mu \in \mathcal{P}_{\leq \ell}} \partial_\mu z_a^{(\ell)} \Delta\mu$, expand the square, and take expectation over $y$ using $\mathbb{E}[y_t] = 0$ and $\mathbb{E}[y_t y_s] = \sigma_y^2 \mathbf{1}\{t = s\}$, independent of the network. The diagonal label contribution produces the $B$-term with $T_{L+1}$, and the remaining terms produce the $A$-term with $S_{L+1}$. $\square$

**Remark** (Top-layer reduction inside the $B$-term). *By Lemma 1 with $h = L+1$, for $\mu_1, \mu_2 \in \mathcal{P}_{\leq \ell}$ we have $\mathbb{E}[T_{L+1}(\mu_1, \mu_2) \mid z^{(\ell)}] = T_\ell(\mu_1, \mu_2)$. Thus $B_{\text{cnn}}^{(\ell)}$ can be rewritten in terms of the layer-$\ell$ overlaps, which is the key step for extracting depth scaling.*

**B.3 From overlap counting to the $-3/2$ depth exponent**

**Negligibility of the $A$-term for the depth exponent.** The term $A_{cnn}^{(\ell)}$ does not alter the depth exponent because it grows at most quadratically in $\ell$ under standard weak-correlation/diagonal-dominance bounds (the same heuristic used in the sequential analysis). Concretely, under homogeneity and finite-moment assumptions at initialization,

$$A_{cnn}^{(\ell)} = O(\eta^2 \ell^2) \quad \text{while} \quad B_{cnn}^{(\ell)} = \Theta(\eta^2 \ell^3), \qquad (21)$$

so averaging over $\ell = 1, \ldots, L$ the $B$-term dominates and sets the depth exponent. We therefore focus on $B_{cnn}^{(\ell)}$.

**Overlap counting.** Using Lemma 1, homogeneity (stationarity under circular padding), and the same overlap-counting argument as in the sequential case (adapted to the effective width $M_\ell = C_\ell N_\ell$), one obtains

$$B_{cnn}^{(\ell)} = \kappa_0 \, \eta^2 \sum_{h_1=1}^{\ell} \sum_{h_2=1}^{\ell} \min\{h_1, h_2\} \cdot \left(1 + O(\max_{r \le \ell} \tfrac{1}{C_r})\right), \qquad (22)$$

where $\kappa_0 = O(1)$ depends on activation/initialization moments (including $q$) but not on $\ell$ or $L$.

The combinatorial identity

$$\sum_{h_1=1}^{\ell} \sum_{h_2=1}^{\ell} \min\{h_1, h_2\} = \frac{\ell(\ell+1)(2\ell+1)}{6} = \Theta(\ell^3) \qquad (23)$$

implies $B_{cnn}^{(\ell)} = \Theta(\eta^2 \ell^3)$.

**Averaging over depth and solving for $\eta_\star(L)$.** Combining Eq. equation 18, Eq. equation 21, and Eq. equation 23, we obtain

$$S_\ell = \mathbb{E}\left[\frac{1}{M_\ell} \sum_{a=1}^{M_\ell} (\Delta z_a^{(\ell)})^2\right] = \kappa_1 \, \eta^2 \, \ell^3 \left(1 + o(1)\right),$$

and therefore

$$\bar{S} = \frac{1}{L} \sum_{\ell=1}^{L} S_\ell = \kappa_2 \, \eta^2 \, L^3 \left(1 + o(1)\right),$$

for constants $\kappa_1, \kappa_2 = O(1)$. Imposing the AM-$\mu$P maximal-update constraint $\bar{S} = \Theta(1)$ yields

$$\eta_\star(L) = \kappa \, L^{-3/2} \left(1 + o(1)\right),$$

with $\kappa = O(1)$ independent of $L$. This proves the depth exponent $-3/2$ in Proposition 3.3. Boundary effects under zero padding contribute only multiplicative corrections as in Remark B, and do not change the exponent.

**2D case.** The 2D CNN proof is identical after replacing the 1D index set by $\Lambda_\ell \subset \mathbb{Z}^2$ and the kernel offset set by $\mathcal{K}_\ell \subset \mathbb{Z}^2$. Under circular padding, each spatial site is again visited exactly $k_\ell := |\mathcal{K}_\ell|$ times, so the kernel-size cancellation and the overlap counting remain unchanged. Under zero padding, the correction is controlled by $\mathrm{bdry}(\Lambda_\ell, \mathcal{K}_\ell)$ as stated in Sec. 3.3.

**Lemma 3** (Boundary missing-term bound for zero padding). *Let $\Lambda \subset \mathbb{Z}^d$ be a finite spatial index set and $\mathcal{K} \subset \mathbb{Z}^d$ be a kernel-offset set with $k := |\mathcal{K}|$. Define the $\mathcal{K}$-boundary set*

$$\partial_{\mathcal{K}} \Lambda := \{ p \in \Lambda : \exists \Delta \in \mathcal{K} \ s.t. \ p + \Delta \notin \Lambda \},$$

$$\mathrm{bdry}(\Lambda, \mathcal{K}) := \frac{|\partial_{\mathcal{K}} \Lambda|}{|\Lambda|}.$$

*Extend any array $f : \Lambda \to \mathbb{R}$ by zero outside $\Lambda$. Then*

$$\left| \frac{1}{|\Lambda|} \sum_{p \in \Lambda} \sum_{\Delta \in \mathcal{K}} f(p + \Delta) - \frac{k}{|\Lambda|} \sum_{u \in \Lambda} f(u) \right| \le \qquad (24)$$

$$k \cdot \mathrm{bdry}(\Lambda, \mathcal{K}) \cdot \|f\|_\infty.$$

*In particular, under circular padding the left-hand side is $0$.*

*Proof.* Under circular padding, each pair $(p, \Delta) \in \Lambda \times \mathcal{K}$ maps to an in-domain index, and each $u \in \Lambda$ is hit exactly $k$ times, so the equality holds exactly.

Under zero padding (with the zero-extension convention), the only discrepancy comes from *missing* terms for which $p \in \partial_{\mathcal{K}} \Lambda$ and $p + \Delta \notin \Lambda$, in which case $f(p+\Delta) = 0$. Each boundary site $p \in \partial_{\mathcal{K}} \Lambda$ can miss at most $k$ offsets. Hence the total number of missing summands is at most $k|\partial_{\mathcal{K}} \Lambda|$, and each missing summand has magnitude at most $\|f\|_\infty$. Dividing by $|\Lambda|$ gives equation 24. $\square$

**Lemma 4** (Mini-batch averaging: $O(1/B)$ correction in second moments). *Let $\{(\xi_b, \zeta_b)\}_{b=1}^{B}$ be i.i.d. pairs with finite second moments. Define the mini-batch averages $\bar{\xi} := \frac{1}{B} \sum_{b=1}^{B} \xi_b$ and $\bar{\zeta} := \frac{1}{B} \sum_{b=1}^{B} \zeta_b$. Then*

$$\mathbb{E}[\bar{\xi} \bar{\zeta}] = \mathbb{E}[\xi_1]\mathbb{E}[\zeta_1] + \frac{1}{B} \mathrm{Cov}(\xi_1, \zeta_1). \qquad (25)$$

*In particular, if $\mathrm{Cov}(\xi_1, \zeta_1) = O(1)$, then $\mathbb{E}[\bar{\xi} \bar{\zeta}]$ differs from the $B = \infty$ limit by $O(1/B)$.*

*Proof.* Expand

$$\mathbb{E}[\bar{\xi} \bar{\zeta}] = \frac{1}{B^2} \sum_{b,b'=1}^{B} \mathbb{E}[\xi_b \zeta_{b'}].$$

For $b \neq b'$, independence gives $\mathbb{E}[\xi_b \zeta_{b'}] = \mathbb{E}[\xi_1]\mathbb{E}[\zeta_1]$. For $b = b'$, $\mathbb{E}[\xi_b \zeta_b] = \mathbb{E}[\xi_1 \zeta_1]$. Thus

$$\mathbb{E}[\bar{\xi}\,\bar{\zeta}] = \frac{1}{B^2}\Big(B\,\mathbb{E}[\xi_1 \zeta_1] + B(B-1)\,\mathbb{E}[\xi_1]\mathbb{E}[\zeta_1]\Big)$$

$$= \mathbb{E}[\xi_1]\mathbb{E}[\zeta_1] + \frac{1}{B}\Big(\mathbb{E}[\xi_1 \zeta_1] - \mathbb{E}[\xi_1]\mathbb{E}[\zeta_1]\Big).$$

which is equation 25. $\qquad\square$

**Remark.** *The $O(1/B)$ correction is stated at the level of second moments of mini-batch averages. The interpretation as a multiplicative factor $1 + O(1/B)$ assumes that the $B \to \infty$ contribution to the one-step AM-$\mu$P budget at initialization is non-vanishing. In degenerate settings where this leading contribution vanishes (e.g., a zero mean-gradient term at initialization), the variance term can dominate, and the learning-rate prefactor may acquire a $\sqrt{B}$ dependence. This does not affect the depth exponent derived in Proposition 3.3.*

**Lemma 5** (Finite-channel correction for $T_h^2$). *Fix a layer $h \in \{1, \dots, L\}$ and two parameter directions $\mu_1, \mu_2$ whose associated parameters are not in layer $h$. Let*

$$T_h(\mu_1, \mu_2) := \frac{1}{C_h |\Lambda_h|} \sum_{j=1}^{C_h} \sum_{p \in \Lambda_h} \partial_{\mu_1} z_{j,p}^{(h)}\, \partial_{\mu_2} z_{j,p}^{(h)},$$

*as in Lemma 1. Condition on the sigma-algebra $\mathcal{F}_{h-1}$ generated by all weights up to layer $h-1$, so that $z^{(h-1)}$ and $\partial_{\mu_s} z^{(h-1)}$ are fixed under the conditional expectation.*

*Assume the weights in layer $h$ are independent across output channels $j$ and have zero mean, and assume the channel-wise second moments are uniformly bounded at initialization:*

$$\mathbb{E}\big[X_1^2 \,\big|\, \mathcal{F}_{h-1}\big] \leq C_0 \qquad \textit{a.s. for some constant } C_0 < \infty, \tag{26}$$

*where*

$$X_j := \frac{1}{|\Lambda_h|} \sum_{p \in \Lambda_h} \partial_{\mu_1} z_{j,p}^{(h)}\, \partial_{\mu_2} z_{j,p}^{(h)}.$$

*Then*

$$\mathbb{E}\big[T_h(\mu_1, \mu_2)^2 \,\big|\, \mathcal{F}_{h-1}\big] = \Big(\mathbb{E}[T_h(\mu_1, \mu_2)\,|\,\mathcal{F}_{h-1}]\Big)^2 + O\Big(\frac{1}{C_h}\Big). \tag{27}$$

*where the implicit constant depends only on $C_0$ (and not on depth). Consequently,*

$$\mathbb{E}\big[T_h(\mu_1, \mu_2)^2\big] = \mathbb{E}\Big[\Big(\mathbb{E}[T_h(\mu_1, \mu_2)\,|\,\mathcal{F}_{h-1}]\Big)^2\Big] + O\Big(\frac{1}{C_h}\Big).$$

*Proof.* Given $\mathcal{F}_{h-1}$, the only randomness comes from the weights in layer $h$. By independence across output channels, the random variables $\{X_j\}_{j=1}^{C_h}$ are i.i.d. conditional

on $\mathcal{F}_{h-1}$. Since $T_h = \frac{1}{C_h} \sum_{j=1}^{C_h} X_j$, we have

$$\mathrm{Var}(T_h \,|\, \mathcal{F}_{h-1}) = \frac{1}{C_h}\, \mathrm{Var}(X_1 \,|\, \mathcal{F}_{h-1})$$

$$\leq \frac{1}{C_h}\, \mathbb{E}\big[X_1^2 \,\big|\, \mathcal{F}_{h-1}\big]$$

$$\leq \frac{C_0}{C_h}.$$

Therefore

$$\mathbb{E}\big[T_h^2 \,\big|\, \mathcal{F}_{h-1}\big] = \Big(\mathbb{E}[T_h \,|\, \mathcal{F}_{h-1}]\Big)^2 + \mathrm{Var}(T_h \,|\, \mathcal{F}_{h-1})$$

$$= \Big(\mathbb{E}[T_h \,|\, \mathcal{F}_{h-1}]\Big)^2 + O\Big(\frac{1}{C_h}\Big).$$

which is equation 27. Taking expectations over $\mathcal{F}_{h-1}$ gives the final displayed statement. $\qquad\square$

**Completion of the proof of Proposition 3.3.** Lemma 3 yields the $\max_\ell \mathrm{bdry}(\Lambda_\ell, \mathcal{K}_\ell)$ correction under zero padding, Lemma 4 yields the $1/B$ correction for mini-batch gradients, and Lemma 5 yields the $\max_\ell 1/C_\ell$ correction from finite-channel averaging. Combined with the overlap-counting derivation in Sec. B, these establish the correction factor $\delta_{\mathrm{cnn}}$ in Proposition 3.3.

## C. Proof of Scaling Law for ResNets

We prove the depthwise learning-rate scaling for residual networks under the same one-step-at-initialization regime used in Appendix B. Throughout this appendix, we consider ReLU activations unless stated otherwise, and we write $q := \mathbb{E}[\sigma'(Z)^2]$ for $Z \sim \mathcal{N}(0,1)$ (so $q = \frac{1}{2}$ for ReLU).

**A simplified homogeneous residual model.** We first analyze a homogeneous residual network with $K$ residual blocks and width $n$. For $\ell = 1, \dots, K$, the $\ell$-th block is

$$z^{(\ell)} = z^{(\ell-1)} + W^{(\ell)} \sigma\big(z^{(\ell-1)}\big), \tag{28}$$

where $z^{(\ell)} \in \mathbb{R}^n$ and each $W^{(\ell)} \in \mathbb{R}^{n \times n}$ has independent entries with

$$\mathbb{E}\Big[W_{ik}^{(\ell)}\Big] = 0, \qquad \mathrm{Var}\Big(W_{ik}^{(\ell)}\Big) = \frac{c}{Kn}, \tag{29}$$

for a constant $c = O(1)$. This matches the residual-branch scaling rule in Sec. 3.1 (up to fixed constants such as $1/q$).

### C.1 Residual-block recursion for $T_\ell$

For a parameter direction $\mu$, we write the directional derivative of the pre-activation at depth $\ell$ as $\partial_\mu z^{(\ell)} \in \mathbb{R}^n$. For two

parameter directions $\mu_1, \mu_2$, define

$$
\begin{aligned}
T_\ell(\mu_1, \mu_2) &:= \frac{1}{n} \left\langle \partial_{\mu_1} z^{(\ell)}, \partial_{\mu_2} z^{(\ell)} \right\rangle \\
&= \frac{1}{n} \sum_{i=1}^{n} \partial_{\mu_1} z_i^{(\ell)} \, \partial_{\mu_2} z_i^{(\ell)}.
\end{aligned}
\tag{30}
$$

**Lemma 6** (One-block conditional recursion for $T_\ell$). *Fix $\ell \in \{1, \ldots, K\}$ and two parameter directions $\mu_1, \mu_2$ whose associated parameters are* not *in block $\ell$ (equivalently, their layer indices are $< \ell$). Condition on the sigma-algebra $\mathcal{F}_{\ell-1}$ generated by all weights up to block $\ell - 1$, so that $z^{(\ell-1)}$ and $\partial_{\mu_s} z^{(\ell-1)}$ are fixed under the conditional expectation. Define the diagonal gate matrix*

$$
D^{(\ell-1)} := \mathrm{diag}\big(\sigma'(z^{(\ell-1)})\big).
$$

*Then we have the exact conditional identity*

$$
\mathbb{E}[T_\ell(\mu_1, \mu_2) \,|\, \mathcal{F}_{\ell-1}] = T_{\ell-1}(\mu_1, \mu_2) + \frac{c}{K} \widetilde{T}_{\ell-1}(\mu_1, \mu_2),
\tag{31}
$$

*where*

$$
\begin{aligned}
\widetilde{T}_{\ell-1}(\mu_1, \mu_2) &:= \frac{1}{n} \left\langle D^{(\ell-1)} \partial_{\mu_1} z^{(\ell-1)}, \, D^{(\ell-1)} \partial_{\mu_2} z^{(\ell-1)} \right\rangle \\
&= \frac{1}{n} \sum_{k=1}^{n} \sigma'\big(z_k^{(\ell-1)}\big)^2 \, \partial_{\mu_1} z_k^{(\ell-1)} \, \partial_{\mu_2} z_k^{(\ell-1)}.
\end{aligned}
\tag{32}
$$

*Proof.* For $s \in \{1, 2\}$, since $\mu_s$ is not in block $\ell$, equation 28 yields

$$
\partial_{\mu_s} z^{(\ell)} = \partial_{\mu_s} z^{(\ell-1)} + W^{(\ell)} D^{(\ell-1)} \partial_{\mu_s} z^{(\ell-1)}.
$$

Let $a^{(s)} := D^{(\ell-1)} \partial_{\mu_s} z^{(\ell-1)} \in \mathbb{R}^n$. Then

$$
\partial_{\mu_s} z^{(\ell)} = \partial_{\mu_s} z^{(\ell-1)} + W^{(\ell)} a^{(s)}.
$$

Plugging into equation 30 and expanding gives

$$
\begin{aligned}
T_\ell(\mu_1, \mu_2) = {}& \frac{1}{n} \left\langle \partial_{\mu_1} z^{(\ell-1)}, \partial_{\mu_2} z^{(\ell-1)} \right\rangle \\
&+ \frac{1}{n} \left\langle \partial_{\mu_1} z^{(\ell-1)}, W^{(\ell)} a^{(2)} \right\rangle \\
&+ \frac{1}{n} \left\langle W^{(\ell)} a^{(1)}, \partial_{\mu_2} z^{(\ell-1)} \right\rangle \\
&+ \frac{1}{n} \left\langle W^{(\ell)} a^{(1)}, W^{(\ell)} a^{(2)} \right\rangle.
\end{aligned}
\tag{33}
$$

Taking conditional expectation given $\mathcal{F}_{\ell-1}$: the first term is exactly $T_{\ell-1}(\mu_1, \mu_2)$. The two middle terms are linear in $W^{(\ell)}$ and vanish since $\mathbb{E}[W_{ik}^{(\ell)}] = 0$. For the last term,

using independence across $(i, k)$ and equation 29,

$$
\begin{aligned}
&\mathbb{E}\left[ \frac{1}{n} \left\langle W^{(\ell)} a^{(1)}, W^{(\ell)} a^{(2)} \right\rangle \,\middle|\, \mathcal{F}_{\ell-1} \right] \\
&= \frac{1}{n} \sum_{i=1}^{n} \mathbb{E}\left[ \left( \sum_{k=1}^{n} W_{ik}^{(\ell)} a_k^{(1)} \right) \left( \sum_{k'=1}^{n} W_{ik'}^{(\ell)} a_{k'}^{(2)} \right) \,\middle|\, \mathcal{F}_{\ell-1} \right] \\
&= \frac{1}{n} \sum_{i=1}^{n} \sum_{k=1}^{n} \mathrm{Var}\big(W_{ik}^{(\ell)}\big) \, a_k^{(1)} a_k^{(2)} \\
&= \frac{c}{K} \cdot \frac{1}{n} \sum_{k=1}^{n} a_k^{(1)} a_k^{(2)} \\
&= \frac{c}{K} \widetilde{T}_{\ell-1}(\mu_1, \mu_2).
\end{aligned}
$$

which together with equation 33 yields equation 31. $\qquad\square$

**Remark.** *To obtain a closed recursion for $\mathbb{E}[T_\ell]$, we use the same mean-field approximation as in Appendix B: at initialization, the gate $\sigma'(z_k^{(\ell-1)})^2$ is asymptotically independent of the product $\partial_{\mu_1} z_k^{(\ell-1)} \partial_{\mu_2} z_k^{(\ell-1)}$. Define the deviation*

$$
\varepsilon_{\ell-1}(\mu_1, \mu_2) := \mathbb{E}\left[ \widetilde{T}_{\ell-1}(\mu_1, \mu_2) \right] - q \, \mathbb{E}[T_{\ell-1}(\mu_1, \mu_2)],
$$

*which satisfies $\varepsilon_{\ell-1}(\mu_1, \mu_2) = o(1)$ as width $\to \infty$ under the same approximation. Taking expectation in equation 31 yields*

$$
\mathbb{E}[T_\ell(\mu_1, \mu_2)] = \left(1 + \frac{cq}{K}\right) \mathbb{E}[T_{\ell-1}(\mu_1, \mu_2)] + \frac{c}{K} \varepsilon_{\ell-1}(\mu_1, \mu_2).
\tag{34}
$$

*For ReLU, $q = \frac{1}{2}$.*

**Corollary.** *Under the same approximation as in Remark C, define*

$$
r_\ell := \left(1 + \frac{cq}{K}\right)^\ell.
$$

*Then for any $\ell \in \{0, 1, \ldots, K\}$,*

$$
\mathbb{E}[T_\ell(\mu_1, \mu_2)] = r_\ell \, \mathbb{E}[T_0(\mu_1, \mu_2)] + \frac{c}{K} \sum_{t=0}^{\ell-1} r_{\ell-1-t} \, \varepsilon_t(\mu_1, \mu_2),
\tag{35}
$$

*where $\varepsilon_t$ is defined in Remark C. Moreover, $r_\ell$ is uniformly bounded in $K$:*

$$
1 \leq r_\ell \leq r_K = \left(1 + \frac{cq}{K}\right)^K \leq e^{cq}.
$$

*In particular, if $\sup_{t \leq K} |\varepsilon_t(\mu_1, \mu_2)| = o(1)$ as width $\to \infty$, then the error term in equation 35 is also $o(1)$ uniformly for $\ell \leq K$.*

## C.2 Depth scaling

**Corollary.** *Consider the residual network equation 28–equation 29 with $K$ residual blocks (each block counts as*

*one depth unit). Under the same one-step update decomposition and weak-dependence assumptions used in Appendix B, and using the $O(1)$ propagation control in Corollary C, the AM-$\mu$P maximal-update learning rate satisfies*

$$\eta_\star(K) = \Theta\big(K^{-3/2}\big).$$

*Proof.* Appendix B shows that for sequential (and homogeneous convolutional) models, the dominant contribution to the one-step pre-activation update second moment can be written in terms of $T_\ell(\mu_1, \mu_2)^2$ and yields the overlap-counting growth

$$\frac{1}{K} \sum_{\ell=1}^{K} \mathbb{E}\Big[(\Delta z^{(\ell)})^2\Big] = \Theta\big(\eta^2 K^3\big).$$

For residual networks, the same decomposition applies. Corollary C shows that the propagation of $T_\ell$ across a residual block differs from the sequential case only by an $O(1)$ factor uniformly in $\ell \leq K$, which affects only the prefactor in the $\Theta(\cdot)$ scaling and does not change the depth exponent. Therefore,

$$\frac{1}{K} \sum_{\ell=1}^{K} \mathbb{E}\Big[(\Delta z^{(\ell)})^2\Big] = \Theta\big(\eta^2 K^3\big),$$

and enforcing the AM-$\mu$P budget $\bar{S} = O(1)$ yields $\eta_\star(K) = \Theta(K^{-3/2})$. $\qquad\square$

**Extension to mixed backbones.** If a backbone contains $m$ plain depth units and $K$ residual blocks along the minimal path (as in Fig. 2), then the same argument applies with $K$ replaced by the effective depth $L_{\text{res}} := m + K$, since plain units satisfy the layerwise invariances proved in Appendix B and residual units contribute only $O(1)$ propagation factors across depth.

# D. Proof of Scaling Law for Transformers

We prove Proposition 3.5. We follow the same first-step analysis as in Appendix B and the same residual-unit recursion logic as in Appendix C; with LayerNorm being the only new technical component.

### D.1 Transformer definition of $T_\ell$

We use the same quantity as in Appendix B–C. Let $z^{(\ell)} \in \mathbb{R}^n$ denote the (flattened) representation after depth unit $\ell$. In Transformers, one may take $n = Nd$ where $N$ is the number of tokens and $d$ is the model width. For two parameter directions $\mu_1, \mu_2$, define

$$T_\ell(\mu_1, \mu_2) := \frac{1}{n} \sum_{i=1}^{n} \partial_{\mu_1} z_i^{(\ell)} \, \partial_{\mu_2} z_i^{(\ell)}. \qquad (36)$$

### D.2 LayerNorm Jacobian bounds

We consider the standard LayerNorm(Ba et al., 2016) map on a $d$-dimensional vector. Let $\bar{x} := \frac{1}{d}\mathbf{1}^\top x$ and

$$s(x) := \sqrt{\frac{1}{d}\|x - \bar{x}\,\mathbf{1}\|_2^2 + \epsilon}, \qquad \epsilon > 0.$$

With affine parameters $\gamma, \beta \in \mathbb{R}^d$, define

$$\text{LN}(x) = \gamma \odot \frac{x - \bar{x}\,\mathbf{1}}{s(x)} + \beta. \qquad (37)$$

We write $\|A\|_{\text{op}} := \sup_{\|v\|_2=1} \|Av\|_2$ for the operator (spectral) norm.

**Lemma 7** (Jacobian of LayerNorm and an operator-norm bound). *Let $J_{\text{LN}}(x)$ denote the Jacobian of equation 37. Then*

$$J_{\text{LN}}(x) = \text{diag}(\gamma) \frac{1}{s(x)} \left( I - \frac{1}{d}\mathbf{1}\mathbf{1}^\top \right.$$
$$\left. - \frac{(x - \bar{x}\,\mathbf{1})(x - \bar{x}\,\mathbf{1})^\top}{d\,s(x)^2} \right). \qquad (38)$$

*Moreover, if $\|\gamma\|_\infty \leq \gamma_{\max}$, then*

$$\|J_{\text{LN}}(x)\|_{\text{op}} \leq \frac{2\,\gamma_{\max}}{s(x)}. \qquad (39)$$

*Proof.* Write $P := I - \frac{1}{d}\mathbf{1}\mathbf{1}^\top$ (centering projection), so $Px = x - \bar{x}\,\mathbf{1}$. Then $\text{LN}(x) = \gamma \odot \big(Px/s(x)\big) + \beta$. Differentiating gives

$$J_{\text{LN}}(x) = \text{diag}(\gamma) \left( \frac{P}{s(x)} + Px \cdot \nabla\left(\frac{1}{s(x)}\right)^\top \right).$$

Since $s(x) = \sqrt{\frac{1}{d}\|Px\|_2^2 + \epsilon}$, we have

$$\nabla\left(\frac{1}{s(x)}\right) = -\frac{1}{s(x)^3} \cdot \frac{1}{d}\,Px.$$

Substituting yields equation 38.

For equation 39, note that $P$ is an orthogonal projection, so $\|P\|_{\text{op}} = 1$. The rank-one matrix $\frac{(Px)(Px)^\top}{d\,s(x)^2}$ has operator norm $\frac{\|Px\|_2^2}{d\,s(x)^2} \leq 1$ by the definition of $s(x)$. Hence the bracketed matrix in equation 38 has operator norm at most 2. Finally, $\|\text{diag}(\gamma)\|_{\text{op}} = \|\gamma\|_\infty \leq \gamma_{\max}$, which gives equation 39. $\qquad\square$

### D.3 A residual depth unit with LayerNorm

We analyze post-norm depth units; the pre-norm case is analogous. A depth unit is written as

$$z^{(\ell)} = \text{LN}\Big(z^{(\ell-1)} + F_\ell(z^{(\ell-1)})\Big), \qquad \ell = 1, \ldots, L_{\text{tr}}, \qquad (40)$$

where $F_\ell$ is a residual branch (either attention or feed-forward), and has $O(1)$ internal depth. Define $u^{(\ell)} := z^{(\ell-1)} + F_\ell(z^{(\ell-1)})$. For notational convenience, write the post-norm depth-unit map as

$$G_\ell(z) := \mathrm{LN}\big(z + F_\ell(z)\big).$$

For $0 \le h < \ell \le L_{\mathrm{tr}}$, define the input-output Jacobian from depth $h$ to depth $\ell$ by

$$\Phi_{\ell:h} := DG_\ell(z^{(\ell-1)})DG_{\ell-1}(z^{(\ell-2)})\cdots$$
$$DG_{h+1}(z^{(h)}), \qquad \Phi_{h:h} := I. \tag{41}$$

**A structural assumption on LayerNorm-stabilized propagation.** The following assumption summarizes the same mean-field / weak-dependence closure used throughout Appendix B–C, now applied to the LayerNorm-stabilized Transformer depth units. It is stated directly for the tangent directions that enter the one-step update expansion.

**Assumption 1** (LayerNorm-stabilized tangent propagation at initialization). *Fix $0 \le h \le \ell \le L_{\mathrm{tr}}$ and condition on $\mathcal{F}_h$, the sigma-algebra generated by all parameters up to depth unit $h$. For any two parameter directions $\mu_1, \mu_2 \in \mathcal{P}_{\le h}$, set*

$$v_s := \partial_{\mu_s} z^{(h)}, \qquad s = 1, 2.$$

*We assume that*

$$\mathbb{E}\left[\frac{1}{n}\langle \Phi_{\ell:h}v_1, \, \Phi_{\ell:h}v_2\rangle \,\Big|\, \mathcal{F}_h\right] = \rho_{\ell:h}\frac{1}{n}\langle v_1, v_2\rangle$$
$$+ r_{\ell:h}(v_1, v_2). \tag{42}$$

*where $\rho_{\ell:h}$ is $\mathcal{F}_h$-measurable and satisfies*

$$0 < c_\Phi \le \rho_{\ell:h} \le C_\Phi < \infty$$

*with constants independent of $L_{\mathrm{tr}}$, $h$, and $\ell$. The remainder satisfies*

$$|r_{\ell:h}(v_1, v_2)| \le \varepsilon_n \frac{\|v_1\|\,\|v_2\|}{n}, \qquad \varepsilon_n \to 0 \tag{43}$$

*as the model width grows. We also assume the corresponding overlap fluctuations are negligible:*

$$\mathrm{Var}\left(\frac{1}{n}\langle \Phi_{\ell:h}v_1, \, \Phi_{\ell:h}v_2\rangle \,\Big|\, \mathcal{F}_h\right) \le \varepsilon_n \frac{\|v_1\|^2\|v_2\|^2}{n^2}. \tag{44}$$

*The constants may depend on the fixed Transformer-block template, the number of heads, the activation function, LayerNorm parameters, and sequence length, but not on the effective depth $L_{\mathrm{tr}}$.*

**Why the assumption is reasonable.** Assumption 1 is the Transformer analogue of the propagation-control conditions used for CNNs and ResNets. LayerNorm keeps tokenwise feature scales $O(1)$ at initialization, while fan-in initialized attention and feedforward projections are approximately isotropic in the large-width mean-field regime. Although the LayerNorm Jacobian has low-dimensional null directions, the parameter-derivative tangent directions appearing in the one-step expansion are high-dimensional random directions, and their projection onto these special directions is negligible at large width. The assumption therefore rules out exponential growth or decay of tangent overlaps with depth, which is precisely the stable Transformer regime considered in the experiments.

**Proposition** (One-unit propagation of $T_\ell$ in post-norm Transformers). *Fix $\ell \in \{1, \ldots, L_{\mathrm{tr}}\}$ and two parameter directions $\mu_1, \mu_2$ whose associated parameters are* not *in depth unit $\ell$. Assume Assumption 1. Then*

$$\mathbb{E}[T_\ell(\mu_1, \mu_2)\,|\,\mathcal{F}_{\ell-1}] = \rho_{\ell:\ell-1}\,T_{\ell-1}(\mu_1, \mu_2)$$
$$+ r_{\ell:\ell-1}\Big(\partial_{\mu_1}z^{(\ell-1)}, \partial_{\mu_2}z^{(\ell-1)}\Big). \tag{45}$$

*where $0 < c_\Phi \le \rho_{\ell:\ell-1} \le C_\Phi < \infty$, with constants independent of $L_{\mathrm{tr}}$, and the remainder is controlled by equation 43. In particular, up to the vanishing remainder, one Transformer depth unit changes the relevant tangent overlap only by a depth-independent constant factor.*

*Proof.* Differentiate equation 40. Since $\mu_s$ is not in depth unit $\ell$,

$$\partial_{\mu_s}z^{(\ell)} = DG_\ell(z^{(\ell-1)})\,\partial_{\mu_s}z^{(\ell-1)} = \Phi_{\ell:\ell-1}\,\partial_{\mu_s}z^{(\ell-1)}.$$

Plugging this identity into equation 36 gives

$$T_\ell(\mu_1, \mu_2) = \frac{1}{n}\left\langle \Phi_{\ell:\ell-1}\partial_{\mu_1}z^{(\ell-1)}, \Phi_{\ell:\ell-1}\partial_{\mu_2}z^{(\ell-1)}\right\rangle.$$

Applying Assumption 1 with $h = \ell - 1$, $v_s = \partial_{\mu_s}z^{(\ell-1)}$, $s = 1, 2$, yields

$$\mathbb{E}[T_\ell(\mu_1, \mu_2)\,|\,\mathcal{F}_{\ell-1}] = \rho_{\ell:\ell-1}\frac{1}{n}\left\langle \partial_{\mu_1}z^{(\ell-1)}, \partial_{\mu_2}z^{(\ell-1)}\right\rangle$$
$$+ r_{\ell:\ell-1}\Big(\partial_{\mu_1}z^{(\ell-1)}, \partial_{\mu_2}z^{(\ell-1)}\Big).$$

The inner product term is exactly $T_{\ell-1}(\mu_1, \mu_2)$, proving equation 45. $\qquad\square$

### D.4 Depthwise scaling

**Corollary** (Depth exponent for Transformers). *Under Assumption 1 and the one-step maximal-update decomposition used throughout Appendix B–C, the AM-$\mu$P maximal-update learning rate $\eta_\star$ (Sec. 3.2) satisfies*

$$\eta_\star(L_{\mathrm{tr}}) = \Theta\big(L_{\mathrm{tr}}^{-3/2}\big).$$

*Proof.* Appendix B shows that, under the one-step decomposition used throughout, the AM-$\mu$P update budget for sequential depth units is governed by overlap sums of the form

$$\sum_{h_1=1}^{\ell} \sum_{h_2=1}^{\ell} \min\{h_1, h_2\},$$

and hence scales as

$$\bar{S} = \Theta\big(\eta^2 L_{\mathrm{tr}}^3\big)$$

up to architecture-dependent constants.

For Transformers, the corresponding sensitivity terms contain the Jacobian propagation from an insertion depth $h$ to a later depth $\ell$. By Assumption 1, this propagation is controlled by the composed LayerNorm-stabilized Jacobian $\Phi_{\ell:h}$:

$$\mathbb{E}\left[\frac{1}{n}\left\langle \Phi_{\ell:h}v_1,\ \Phi_{\ell:h}v_2\right\rangle \middle| \mathcal{F}_h\right] = \rho_{\ell:h}\frac{1}{n}\langle v_1, v_2\rangle + r_{\ell:h}(v_1, v_2),$$

where

$$0 < c_\Phi \le \rho_{\ell:h} \le C_\Phi < \infty$$

with constants independent of $L_{\mathrm{tr}}$, $h$, and $\ell$, and the remainder is negligible in the large-width limit. Thus, relative to the sequential overlap calculation, Transformer depth units modify the relevant tangent overlaps only by depth-independent constants. The fluctuation bound in Assumption 1 ensures that the same replacement is valid at the second-moment level entering the AM-$\mu$P budget.

Consequently, the same overlap-counting argument gives

$$\bar{S} = \Theta\big(\eta^2 L_{\mathrm{tr}}^3\big)$$

for LayerNorm-stabilized Transformer depth units. Enforcing the AM-$\mu$P budget $\bar{S} = \Theta(1)$ therefore yields

$$\eta_\star(L_{\mathrm{tr}}) = \Theta\big(L_{\mathrm{tr}}^{-3/2}\big).$$

$\square$

**Remark.** *For pre-norm units of the form*

$$z^{(\ell)} = z^{(\ell-1)} + F_\ell(\mathrm{LN}(z^{(\ell-1)})),$$

*the same argument applies after redefining the depth-unit map as*

$$G_\ell(z) := z + F_\ell(\mathrm{LN}(z))$$

*in the definition of $\Phi_{\ell:h}$. In this case, Assumption 1 is imposed on the corresponding pre-norm LayerNorm-stabilized Jacobian propagation. Under this analogous tangent-propagation stability condition, the depth exponent is unchanged.*

# E. Additional Experimental Results

## E.1. Ablation Study Details

Table 3 summarizes all ablation study configurations and their fitted depth exponents. Figure 5 presents the corresponding log-log fitting curves, demonstrating consistent power-law relationships across all tested configurations.

## E.2. CNN Padding Ablation

We compare circular padding and zero padding under identical CNN settings on CIFAR-10 with ReLU activation. Figure 6 shows that both padding modes follow essentially the same depth–learning-rate power law, with exponents close to the theoretical $L^{-3/2}$ prediction. The primary difference manifests as a small vertical shift on the log scale (i.e., a constant prefactor change) rather than a slope change, confirming that padding strategy has minimal impact on the scaling exponent(Xiao et al., 2018). This validates zero padding as a practical default in engineering applications while maintaining the theoretical scaling law.

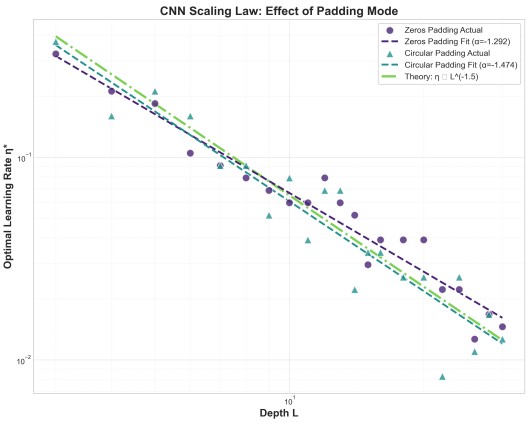

*Figure 6.* **CNN padding comparison on CIFAR-10.** Depth–LR scaling curves for circular padding and zero padding with ReLU activation. Both exhibit similar slopes ($\hat{\alpha} \approx -1.5$), differing primarily in vertical offset.

## E.3. Kernel-Size Ablation

Proposition 3.3 predicts that for stride-1 CNNs under AM-$\mu$P, the maximal-update learning rate follows $\eta^\star(L) = \kappa L^{-3/2}$ at leading order, where kernel size affects the learning rate primarily through the prefactor $\kappa$ rather than the depth exponent. To test this prediction, we repeat the CNN/CIFAR-10 protocol with kernel sizes $k \in \{3, 4, 5\}$ while keeping all other settings fixed.

Figure 7 shows that $\eta^\star$ exhibits clear power-law decay for all kernel sizes, with fitted exponents $\hat{\alpha} \in \{-1.74, -1.44, -1.46\}$. Importantly, varying $k$ does not produce a monotonic trend in $\hat{\alpha}$; instead, it primarily in-

*Table 3.* **Ablation study: depth exponents for representative configurations.** Fitted slopes $\hat{\alpha}$ for the depth–LR scaling law $\log_{10} \eta^{\star} = \beta_0 + \alpha \log_{10} L$. Theory predicts $\alpha = -1.5$. Base configuration uses SGD, ReLU, and no regularization unless otherwise specified.

| Model | Configuration | Optimizer | Dataset | Details | $\hat{\alpha}$ |
|---|---|---|---|---|---|
| CNN | Baseline | SGD | CIFAR-10 | ReLU, None, – | $-1.339$ |
|  | + GELU(Hendrycks, 2016) | SGD | CIFAR-10 | GELU, None, – | $-1.379$ |
|  | + Adam(Kingma & Ba, 2017) | Adam | CIFAR-10 | ReLU, None, – | $-1.207$ |
| ResNet | Baseline | SGD | CIFAR-10 | ReLU, None, – | $-1.435$ |
|  | + BatchNorm(Ioffe & Szegedy, 2015) | SGD | CIFAR-10 | ReLU, BN, – | $-1.701$ |
|  | + Dropout(Srivastava et al., 2014) | SGD | CIFAR-10 | ReLU, None, Dropout | $-1.568$ |
|  | + Adam | Adam | CIFAR-10 | ReLU, None, – | $-1.269$ |
| ViT | Pre-LN | SGD | ImageNet | ReLU, Pre-LN, – | $-1.131$ |
|  | Post-LN | SGD | ImageNet | ReLU, Post-LN, – | $-1.178$ |

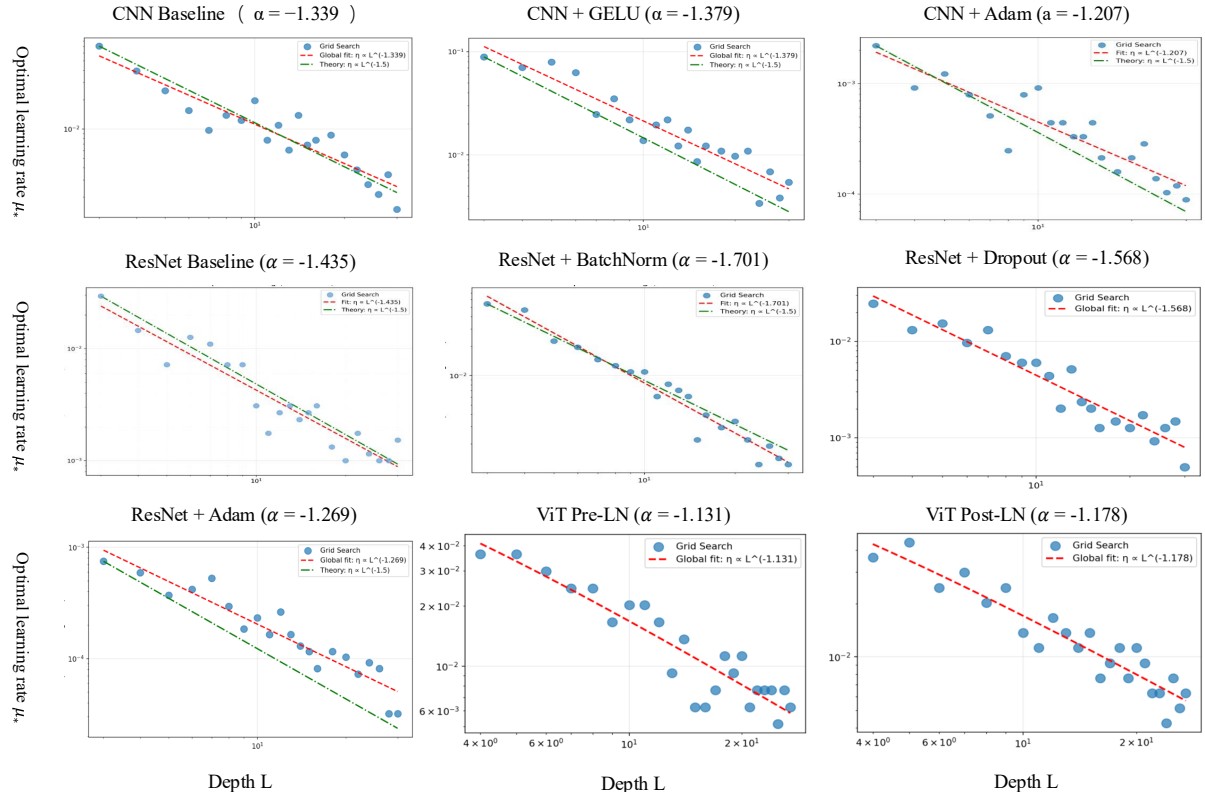

*Figure 5.* **Depth–LR scaling curves for ablation configurations.** Each subplot shows $\log_{10} \eta^{\star}$ versus $\log_{10} L$ with the fitted power-law line. Configurations correspond to Table 3. All configurations exhibit clear power-law relationships with exponents close to the theoretical prediction of $-1.5$.

*Table 4.* ViT/CIFAR-10 width and later-epoch robustness. Each entry is $\hat{\alpha}$ with $R^2$ in parentheses.

| Width | Epoch 1 | Epoch 2 | Epoch 3 |
|---|---|---|---|
| small (288) | $-1.352\,(0.938)$ | $-1.531\,(0.943)$ | $-1.207\,(0.739)$ |
| base (384) | $-1.360\,(0.980)$ | $-1.444\,(0.928)$ | $-1.501\,(0.898)$ |
| large (480) | $-1.345\,(0.990)$ | $-1.647\,(0.961)$ | $-1.550\,(0.924)$ |

*Table 5.* Representative direct zero-shot transfer results on ViT/CIFAR-10. Here $\eta_{\mathrm{or}}$ is the independently tuned oracle LR, $\eta_{\mathrm{sc}}$ is the theory-scaled LR, and log-LR errors are measured in decades. Losses are epoch-3 training losses.

| $D$ | $\eta_{\mathrm{or}}$ | $\eta_{\mathrm{sc}}$ | $e_{\mathrm{raw}}$ | $e_{\mathrm{sc}}$ | raw / sc. loss |
|---|---|---|---|---|---|
| 6 | $5.360\times10^{-3}$ | $6.231\times10^{-3}$ | $0.338$ | $0.065$ | $1.780/1.768$ |
| 8 | $4.874\times10^{-3}$ | $4.274\times10^{-3}$ | $0.297$ | $0.057$ | $1.772/1.770$ |
| 10 | $3.249\times10^{-3}$ | $3.163\times10^{-3}$ | $0.120$ | $0.012$ | $1.785/1.785$ |
| 20 | $1.194\times10^{-3}$ | $1.199\times10^{-3}$ | $0.314$ | $0.002$ | $1.833/1.833$ |

duces a vertical shift (prefactor change) while the exponent remains close to the theoretical $3/2$. We therefore interpret the variation in fitted slopes (within $\pm20\%$ of $3/2$) as finite-sample fluctuations rather than evidence of a kernel-dependent depth exponent, confirming that kernel size has minimal impact on the scaling law.

median log-LR error from $0.314$ to $0.057$ decades and achieves lower unrounded epoch-3 training loss than raw transfer on $6/7$ targets. Table 5 shows representative target depths.

**Preliminary non-vision check.** As a preliminary non-vision sanity check, we run an audio classification sweep. The fitted slope is $\hat{\alpha} = -1.578$ with $R^2 = 0.891$, close to the predicted $-1.5$ trend.

*Table 6.* Preliminary audio classification depth–LR sweep.

| Effective depth $L$ | Best LR |
|---|---|
| 6 | $6.31\times10^{-2}$ |
| 10 | $2.39\times10^{-2}$ |
| 14 | $2.39\times10^{-2}$ |
| 18 | $9.03\times10^{-3}$ |

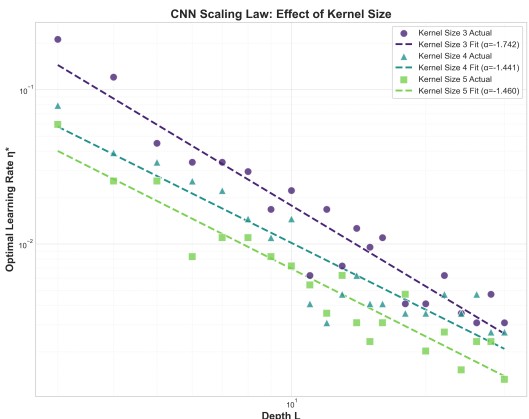

*Figure 7.* **Kernel-size ablation on CIFAR-10.** Maximal-update learning rate $\eta^\star$ versus depth $L$ for kernel sizes $k \in \{3,4,5\}$. Dashed lines show log-log linear fits with exponents close to the theoretical $-1.5$.

### E.4. Additional Transfer Checks

**Proxy width and later epochs.** Table 4 repeats the ViT/CIFAR-10 sweep across three proxy widths and selects the best LR after epochs 1–3. The fitted slopes remain concentrated around the predicted leading exponent $\alpha = -1.5$. The smallest-width epoch-3 fit is noisier, while the base and large settings remain close to the predicted trend.

**Direct zero-shot depth transfer.** We next test the rule as a transfer procedure. A source LR $\eta_0 = 2.462 \times 10^{-3}$ is calibrated at source block depth $D_0 = 12$ and transferred either unchanged or after depth rescaling,

$$\eta_{\mathrm{sc}}(L) = \eta_0 \left(\frac{L}{L_0}\right)^{-3/2},$$

where $L_0$ is the corresponding source effective depth. Oracle LRs are obtained by independent grid search at each target depth under the same three-epoch budget. Across all non-source target depths, the scaled rule reduces the

## F. Why GELU Can Appear Slightly Steeper than ReLU in Finite-Range Fits

Empirically, when we fit a *single* depth–learning-rate power law over a finite depth range, the fitted exponent for GELU can be marginally more negative than for ReLU for the same architecture and training setup. This does not necessarily indicate a different asymptotic exponent. Rather, it suggests that subleading (finite-depth/finite-width) corrections are more pronounced for GELU.

**Activation–derivative statistics and sensitivity.** Let $Z \sim \mathcal{N}(0, 1)$ denote the Gaussian proxy used in the mean-field calculations. For ReLU, $\mathbb{E}[\sigma'(Z)^2] = \frac{1}{2}$. For GELU $\phi(x) = x\Phi(x)$ with $\phi'(x) = \Phi(x) + x\varphi(x)$, we have

$$\mathbb{E}[\phi'(Z)^2] \approx 0.456.$$

If one keeps the ReLU-calibrated He variance $\mathrm{Var}(W) = 2/\mathrm{fan}_{\mathrm{in}}(W)$ when switching from ReLU to GELU, then the linearized backward gain per layer becomes $\chi = 2\,\mathbb{E}[\phi'(Z)^2] \approx 0.912 < 1$, which increases the tendency of deeper layers to attenuate gradients at initialization. When fitted as a *single* power law over a limited depth range, this attenuation can bias the estimated slope.

**Drift at finite depth and width** Even with activation-aware fan-in scaling, the pre-activation distribution is only approximately stationary at finite width and finite depth. For smooth activations such as GELU, the effective quantity $q(z) = \mathbb{E}[\phi'(z)^2]$ is more sensitive to small variance drifts across layers. This induces a slowly depth-dependent prefactor $\kappa(L)$ in $\eta_\star(L) \approx \kappa(L) L^{-3/2}$. A log–log regression that enforces a single power law can therefore return a slightly more negative fitted exponent.

# G. On the Loss: Cross-Entropy vs. MSE in the One-Step Derivation

Our one-step maximal-update derivation uses MSE for analytic convenience, while the main experiments use multi-class cross-entropy (CE)(Janocha & Czarnecki, 2017). In the one-step initialization regime, this choice does not change the depth exponent, because the derivation only requires that the logit-gradient has an $O(1)$ second moment.

At initialization, logits are near zero, so $p = \text{softmax}(z)$ is close to uniform. For one-hot targets $y$ with $C$ classes, the CE logit gradient is

$$g \;=\; p - y,$$

and thus

$$\|g\|_2^2 = \left(1 - \tfrac{1}{C}\right)^2 + (C-1)\left(\tfrac{1}{C}\right)^2 = 1 - \tfrac{1}{C} = O(1).$$

Therefore CE provides $O(1)$-scale per-sample gradients at initialization. In our derivation, the depth dependence of $\eta_\star(L)$ is governed by Jacobian products through the network, while the loss enters only through such $O(1)$ output-gradient statistics. Replacing MSE by CE consequently rescales the overall prefactor $\kappa$ but does not change the depthwise power-law exponent.

