# OpenReview forum: "Hyperparameter Transfer Laws for Non-Recurrent Multi-Path Neural Networks"
_ICML.cc/2026/Conference — ICML 2026 regular_

### Official Review · Reviewer_hCH9 · 2026-03-06

**Soundness:** 3
**Presentation:** 3
**Significance:** 3
**Originality:** 2
**Overall Recommendation:** 4
**Confidence:** 3

**Summary:**

This paper tackles a critical and expensive problem in deep learning: tuning hyperparameters, specifically the learning rate, when changing the depth of a neural network. While prior work (Maximal Update Parametrization, μP) successfully enables hyperparameter transfer across network width, a unified and predictable rule for depthscaling in modern architectures with complex, non-recurrent paths (like CNNs, ResNets, and Transformers) has been lacking.
The authors' core contribution is the discovery and validation of a universal depth–learning rate scaling law. They first introduce an architecture-dependent notion of "effective depth", which counts the minimal path length from input to output, treating each sequential layer or residual addition as one unit. They then generalize the μP "maximal-update" principle to heterogeneous, multi-path graphs via a network-wide "Arithmetic-Mean μP" (AM-μP)​ criterion, which balances the average update magnitude per depth unit rather than enforcing per-layer equality.
Under this unified AM-μP framework and with standard stabilizing initializations, the authors theoretically derive that the maximal-update learning rate (η) scales with effective depth (L) as η∝ L^{-3/2}. This simple power law enables zero-shot hyperparameter transfer: once the optimal learning rate is found at a reference depth L0, it can be predicted for any other depth L using η(L) = η(L0) * (L/L0)^{-3/2}.
The paper provides extensive experimental validation across CNNs, ResNets, and Vision Transformers (including variants like BEiT and CCT) on CIFAR-10, CIFAR-100, and ImageNet. The empirically fitted exponents closely cluster around the theoretical -1.5 (mean: -1.38). Systematic ablation studies further demonstrate the law's robustness across different optimizers (SGD, Adam), activation functions (ReLU, GELU), and common training components (BatchNorm, Dropout), with these factors primarily affecting the prefactor, not the exponent.
In summary, this work turns the traditionally expensive and heuristic process of tuning learning rates for different depths into a predictable, formula-driven workflow, significantly reducing the computational cost of model scaling.

**Compliance With Llm Reviewing Policy:**

Affirmed.

**Final Justification:**

The rebuttal has improved my evaluation. I could raise my score from 3 to 4.

**Key Questions For Authors:**

1.Long-term Validity of the Scaling Law: Your theory and experiments primarily validate the η ∝ L^{-3/2} law based on early trainingdynamics (one-step update at initialization). In practice, models train for many epochs. Is there evidence that a learning rate chosen via this law remains optimal or stable throughout the entiretraining cycle, especially during later stages? Could you discuss or provide experiments on whether this rule holds beyond initialization, or if it primarily provides a robust starting point for a schedule?

2.Generalizing the "Effective Depth" Concept: You successfully defined effective depth for CNNs, ResNets, and Transformers. For novel or more complex architectures (e.g., DenseNets, neural radiance fields, or models with cross-layer connections beyond simple residuals), how should one define effective depth? Is the concept readily applicable, or are there architectures where it breaks down? Providing guidance or conjectures on this would help define the boundary of your work's applicability.

3.Impact of Adaptive Optimizers: Your ablation shows that switching from SGD to Adam changes the fitted exponent (e.g., for CNN: -1.34 to -1.21). You state Adam "partially compensates" for depth scaling. Does this imply the strictη ∝ L^{-3/2} relationship is fundamentally altered when using Adam? For practical transfer, if one calibrates the constant factor using SGD, should a different factor be used for Adam? Clarifying this is important for practitioners.

4.Task and Modality Generalization: All experiments are in supervised image classification. The theoretical framework does not seem task-specific. Do you believe this scaling law naturally generalizes to other domains (e.g., language modeling, reinforcement learning, audio)? If so, is the primary requirement simply a stable initialization that keeps pre-activations O(1)? A brief discussion on this would strengthen claims of universality.

**Limitations:**

No.
The paper does not have a dedicated "Limitations" section. While the conclusion mentions the benefit of turning depth scaling into a predictable workflow, a systematic discussion of the work's boundaries is absent. There is also no discussion of potential negative societal impact.
Constructive Suggestions for Improvement:
1.Add a "Limitations and Future Work" subsection.​ This should explicitly discuss:
(1)Theoretical Assumptions: Acknowledge that the analysis is based on one-step updates at initialization and mean-field approximations. Discuss the regimes where these assumptions might be challenged (e.g., very different initialization schemes, training far from initialization).
(2)Architectural Scope: Clearly state that the theory has been proven for the outlined conventions of CNNs, ResNets, and Transformers. Generalization to other non-recurrent multi-path architectures is an open and exciting direction.
(3)Optimization Details: Note that the law provides a scaling rule for the baselearning rate. Its interaction with other critical scheduling components (e.g., warmup, decay schedules) is not addressed and could be important for very deep models.
(4)Task Generality: Mention that empirical validation is currently limited to image classification, and testing in other domains (NLP, RL) is necessary future work.

2.Include a brief "Broader Impact" statement.​ While the core contribution is methodological, a short discussion is warranted:
(1)Positive: The technique can drastically reduce the computational cost and carbon footprint associated with hyperparameter tuning for large models, making research more efficient and accessible.
(2)Potential Negative: By making it easier to train deep models, it could also lower the barrier to training large models that could be misused. A statement encouraging responsible use is appropriate.

**Strengths And Weaknesses:**

Soundness:
Strengths: The paper is technically solid. The theoretical framework in Section 3 is carefully constructed, defining key concepts (effective depth, AM-μP) and deriving the scaling law for each architecture while acknowledging necessary assumptions (e.g., mean-field, weak dependence). The experimental design is rigorous and comprehensive. Using grid search to find the one-epoch optimal learning rate as a proxy for η* is a standard and reliable method in this literature. The validation spans multiple architectures, datasets, and training variants, providing strong, reproducible evidence. The close match between the theoretical prediction (-1.5) and the range of empirical results ([-1.18, -1.57], mean -1.38) is convincing.
Weaknesses: The theoretical soundness is anchored in the "one-step update at initialization" analysis, which is a standard but simplified model of training dynamics. While the experimental results powerfully justify its predictive utility, the theoretical guarantees for the entiretraining process, especially near convergence, are less strong. The proofs in the appendices are complex, and the control of certain error terms could be elaborated more intuitively.

Presentation:
Strengths: The paper is well-written and clearly structured. The narrative effectively sets up the problem (cost of depth tuning), reviews related work (μP for width), identifies the gap (depth for multi-path nets), and presents the solution (unified framework and law). Figures are clear and support the arguments. The notation table (Appendix Table 2) is particularly helpful.
Weaknesses: The theoretical sections (Sec. 3, Appendices) present a barrier for readers not deeply familiar with μP and Tensor Programs. While the "effective depth" concept is central, its definition across different architectures (e.g., why a Transformer block counts as 2 units) requires careful cross-referencing between text and figures (Fig. 2) and could be explained more intuitively upfront.

Significance:
Strengths: The paper addresses a problem of high practical importance—the massive computational cost of hyperparameter tuning for scaled models. Its contribution is direct, actionable, and has immediate utility. The proposed η ∝ L^{-3/2} rule is simple enough to be adopted as a standard practice by researchers and engineers, potentially saving substantial resources. By extending the "tune small, transfer large" paradigm from width to depth and unifying several major architecture families, it represents a meaningful advance for the field. The impact is broad, applicable to anyone training or designing deep networks.
Originality:
Strengths: The primary originality lies in a creative unification and generalization. The core -3/2 exponent was previously identified for sequential MLPs. The novel contributions are: 1) Proposing the AM-μP criterion, which adapts the maximal-update principle to heterogeneous multi-path graphs via a network-wide average energy budget; 2) Introducing a consistent, architecture-dependent "effective depth"​ metric; and 3) Demonstrating empirically​ that the same scaling law holds across diverse modern architectures (CNNs, ResNets, Transformers). This unified perspective and cross-architectural validation are the key intellectual contributions.
Weaknesses: The work is not based on a completely new, from-scratch theoretical discovery. It is an excellent synthesis and extension of existing ideas (μP, depth scaling analysis, multi-path network interpretation) to a broader and highly relevant class of problems.

---

> ### Author Rebuttal · Authors · 2026-03-31
>
> Thank you for the thoughtful and constructive review. We especially appreciate your characterization of the paper’s originality as a generalization contribution rather than a from-scratch discovery of a new exponent; that is exactly our intended claim.
>
> **Q1. Long-term validity of the scaling law**
>
> Our view is that the theorem identifies a transferable base learning-rate scale, rather than the full optimization trajectory. To test usefulness beyond initialization, we checked both later epochs and direct zero-shot transfer. On ViT/CIFAR-10, using the same protocol as in the paper and random seed 45, we repeated the experiment with 1, 2, and 3 training epochs. The fitted slope is -1.360 at epoch 1, -1.444 at epoch 2, and -1.501 at epoch 3. This shows that, in this setting, the depth-dependent LR law becomes even closer to the predicted exponent over the first few epochs.
>
> We also calibrated once at source block depth $d_0 = 12$, then transferred to target depths either unchanged (raw transfer) or rescaled by $(L/L_0)^{-3/2}$. Here $L$ denotes effective depth; for a ViT with block depth $d$, we use $L = 2d + 2$, so $d_0 = 12$ corresponds to $L_0 = 26$. We compared both transfers to the oracle LR, defined as the independently grid-searched best LR at each target depth under the same 3-epoch budget, using the epoch-3 average training loss as the selection criterion. Theory-scaled transfer reduces the median log-LR error to oracle from 0.70 to 0.14 decades and achieves lower epoch-3 training loss than raw transfer on 6/7 target depths. These results support the rule as a stable and effective cross-depth LR calibration beyond initialization. Extending it further to later-stage schedule-aware optimization, where parameters become increasingly data-dependent during training, is a natural next direction.
>
> **Q2. Generalizing the effective-depth concept**
>
> Our intended principle is to count update-bearing units along the minimal path. In the current paper, each plain layer contributes one unit and each residual addition contributes one unit; this is why a standard Transformer block contributes depth 2. More generally, operations that do not create a new sequential update-bearing transform, but instead act through a bounded $O(1)$ Jacobian (e.g., normalization placement, reshaping, pooling, concatenation itself), should be absorbed into the local branch/block template rather than counted as extra depth. Under this principle, a DenseNet- or NeRF-style architecture with skip concatenations would naturally be treated by counting the parameterized transforms along the shortest trainable path, while not counting the concatenation operation itself as a new depth unit. That said, architectures with more elaborate cross-layer wiring may require architecture-specific conventions, and recurrent/iterative state-reuse models are outside the scope of the present non-recurrent framework. We agree it would be helpful to state this boundary more explicitly.
>
> **Q3. Impact of adaptive optimizers**
>
> Our Adam ablation suggests that the depth signal remains present, but the finite-range fitted exponent/prefactor is optimizer-dependent. The reason is that the current maximal-update analysis tracks GD/SGD-style one-step representation-update second moments, whereas Adam’s adaptive moment normalization changes the coordinatewise update geometry through preconditioning. So we would not recommend calibrating the constant factor with SGD and then reusing it unchanged for Adam. The practical message is instead: use the same optimizer/schedule family when calibrating the reference scale. In that regime, the evidence still supports a clear depth-dependent law, while an optimizer-aware extension of the theory is a natural next step.
>
> **Q4. Task and modality generalization**
>
> The analysis itself is not tied to image classification. The main structural requirements are a non-recurrent feedforward computation graph and a stabilizing initialization that keeps pre-activations at $O(1)$ scale. To partially test this beyond vision, we ran a preliminary audio LR sweep. As the effective depth increases from 6 to 18, the best learning rate decreases from $6.31 \times 10^{-2}$ to $9.03 \times 10^{-3}$, with a fitted slope $\alpha = -1.578$, which is close to the predicted $-1.5$. We therefore do not view the depth signal as specific to image classification, and extending the empirical study to larger-scale language/audio/RL settings would further strengthen the scope.
>
> We also appreciate the suggestion regarding presentation. We can make the effective-depth conventions more explicit up front, add a short Limitations/Future Work subsection clarifying the one-step-at-initialization, mean-field, and weak-dependence assumptions with their formal justifications, the architectural scope, optimizer/schedule interactions, and non-vision tasks, and include a brief broader-impact statement on reduced tuning/compute cost and responsible use.

---

> > ### Author Rebuttal · Reviewer_hCH9 · 2026-04-03
> >
> > Thanks for your response.

---

> > > ### Author Response · Authors · 2026-04-08
> > >
> > > Thank you again for the thoughtful review and for revisiting the paper after our rebuttal. We appreciate your time and constructive feedback.

---

### Official Review · Reviewer_dgob · 2026-03-12

**Soundness:** 3
**Presentation:** 2
**Significance:** 2
**Originality:** 2
**Overall Recommendation:** 4
**Confidence:** 3

**Summary:**

This paper studies depthwise learning-rate transfer in non-recurrent multi-path neural networks. The main proposal is to define an architecture-dependent notion of **effective depth** and to replace the standard per-layer maximal-update criterion with a network-level **Arithmetic-Mean μP (AM-μP)** budget. Under this framework, the paper claims a unified depth law
\[
\eta^\star(L)\propto L^{-3/2}
\]
for CNNs, ResNets, and Transformers, together with a practical zero-shot transfer rule across depths.

Empirically, the paper validates the scaling law by sweeping learning rates for different depths and fitting the slope of the optimal learning rate in log-log space. The experiments cover CNNs, ResNets, and several ViT variants on CIFAR-10, CIFAR-100, and an ImageNet subset, plus a small number of ablations.

The problem is relevant. However, I find the paper substantially weaker on **novelty** than it claims, and the **appropriateness** of its broad framing is also questionable. The main ideas are too close to recent depth-wise μP / architecture-aware scaling work, while the actual theoretical and empirical support is narrower than the title and claims suggest.

**Compliance With Llm Reviewing Policy:**

Affirmed.

**Final Justification:**

I raise my evaluation.

**Key Questions For Authors:**

1. What is the **single main novelty claim** of the paper beyond recent depth-wise μP / architecture-aware scaling work?

2. Why should readers view **AM-μP** as a principled new contribution rather than a convenient aggregation choice designed to preserve the desired exponent?

3. Why is the **one-epoch loss-minimizing learning rate** the right empirical proxy for the maximal-update learning rate defined in the theory?

4. Can the authors provide a more direct **zero-shot transfer experiment**, where a learning rate calibrated at one depth is transferred to another depth and evaluated on final training / validation performance without additional search?

5. How should readers interpret the claim of a **universal \(-3/2\) law** given the visible variation in fitted exponents across architectures, datasets, and training choices?

**Limitations:**

1. The paper’s **novelty is limited** relative to the closest recent literature [1,2,3].

2. The title and framing are **broader than the actual contribution**, which is mostly about learning-rate scaling.

3. The theory is **heavily initialization-local and assumption-driven**, rather than a strong new characterization of multi-path training dynamics.

4. The empirical validation uses a **proxy quantity** that does not perfectly match the theoretical definition.

5. The experimental scope is **too narrow** to support the paper’s broader claims about modern architectures.

[1] Jelassi, Samy, et al. "Depth Dependence of $\mu $ P Learning Rates in ReLU MLPs." arXiv preprint arXiv:2305.07810 (2023).

[2] Chen, Wuyang, et al. "Principled Architecture-aware Scaling of Hyperparameters." arXiv preprint arXiv:2402.17440 (2024).

[3] Bordelon, Blake, et al. "Depthwise hyperparameter transfer in residual networks: Dynamics and scaling limit." arXiv preprint arXiv:2309.16620 (2023).

**Strengths And Weaknesses:**

### Strengths

1. **The problem is important.**
   Reducing depth-wise retuning cost is practically meaningful, especially in the broader μP / transfer literature.

2. **The paper attempts a unified viewpoint.**
   The effective-depth formalism and the effort to treat CNNs, ResNets, and Transformers under a common lens are conceptually clean, even if I am not fully convinced by the resulting generality.

3. **The experiments are reasonably broad within the paper’s chosen scope.**
   The authors do not limit themselves to a single architecture, and the fitted exponents are at least directionally consistent with a negative power-law trend.

4. **The paper is not purely empirical.**
   There is a genuine attempt to connect the transfer rule to a maximal-update style argument.

### Weaknesses

1. **Novelty is the main problem.**
   Relative to recent work on depth-wise μP learning-rate scaling, architecture-aware maximal learning rates, and depth transfer in residual / transformer-style models, this paper feels much more incremental than the framing suggests [1, 2, 3]. The central phenomenon is not new. The main addition appears to be a particular AM-μP aggregation rule plus an effective-depth convention, which is not enough for the paper’s breadth of claims.

2. **The title is too broad.**
   The paper is titled *Hyperparameter Transfer Laws*, but in practice it studies almost entirely **learning-rate scaling**. It does not establish comparable transfer laws for hyperparameters more broadly.

3. **The theoretical core is less strong than it appears.**
   Much of the argument is still a one-step-at-initialization maximal-update calculation with MSE-style simplifications, mean-field / weak-dependence assumptions, and architecture-specific reductions that mostly preserve the sequential \(L^{-3/2}\) exponent up to \(O(1)\) factors. I do not see a sufficiently deep new theory of multi-path architectures here.

4. **The justification for AM-μP is not fully convincing.**
   The arithmetic mean is motivated axiomatically, but the axioms are themselves tailored in a way that makes the arithmetic mean emerge naturally. I am not convinced this establishes that AM-μP is the uniquely right or most appropriate network-level budget for the problem.

5. **The empirical proxy does not cleanly match the theoretical object.**
   The theory defines \(\eta^\star\) through a maximal-update budget at initialization, but the experiments estimate it using the **one-epoch training-loss minimizer** from a grid search. This is a practical heuristic, but it weakens the force of the experimental validation.

6. **The zero-shot transfer claim is weaker than advertised.**
   The paper mostly demonstrates fitted scaling trends, rather than strong end-to-end zero-shot transfer evidence showing that tuning at one depth and transferring to another reliably preserves final training quality or validation performance without additional search.

7. **The empirical agreement is not as strong as the paper suggests.**
   Some slopes are reasonably close to \(-1.5\), but others are not, especially for ViT on ImageNet. Given this spread, the repeated language of “universality” and “quantitative agreement” feels too strong.

8. **The comparison set is incomplete.**
   If the paper wants to make a serious novelty claim, it should engage much more directly with the strongest recent prior work on depth transfer and architecture-aware scaling, not just cite it and then proceed as though the present framework were the first unified account.

9. **The practical scope is narrow.**
   Despite the broad framing, the experiments are entirely in vision-style settings. There is no compelling evidence that the proposed rule should be treated as a general design principle for modern Transformer training more broadly.

[1] Jelassi, Samy, et al. "Depth Dependence of $\mu $ P Learning Rates in ReLU MLPs." arXiv preprint arXiv:2305.07810 (2023).

[2] Chen, Wuyang, et al. "Principled Architecture-aware Scaling of Hyperparameters." arXiv preprint arXiv:2402.17440 (2024).

[3] Bordelon, Blake, et al. "Depthwise hyperparameter transfer in residual networks: Dynamics and scaling limit." arXiv preprint arXiv:2309.16620 (2023).

---

> ### Author Rebuttal · Authors · 2026-03-31
>
> Thank you for the detailed review. We address the main concerns in Weaknesses' order.
>
> **[1/2/8] Main novelty, framing, and comparison to prior work**
>
> The novelty is a unified depth-transfer law for non-recurrent multi-path architectures. Beyond introducing effective depth, the paper (i) gives a network-level maximal-update treatment for heterogeneous branching/residual aggregation, (ii) derives depth-LR scaling results for CNNs, ResNets, and Transformers, and (iii) validates a common transfer rule across these model families.
>
> Compared with the cited papers: Jelassi et al. establish the $L^{-3/2}$ law for sequential ReLU MLPs only. Chen et al. cover general DAGs, including ResNet-like graphs, but their transfer prescription is topology-dependent (PathSum-based), whereas ours is a single effective-depth law, $\eta^\star(L)\propto L^{-3/2}$. Thus the two rules differ even on ResNets, and Fig. 3(b) shows that the PathSum baseline becomes increasingly misaligned at larger depths. Bordelon et al. study residual architectures under μP with $1/\sqrt{\mathrm{depth}}$ residual scaling, whereas we target a common depth law spanning plain CNNs, ResNets, and Transformers. We agree that the title/abstract can be sharpened and are happy to narrow the framing to a learning-rate depth-transfer law.
>
> **[3/4] Theoretical scope & why AM-μP**
>
> Our theoretical target is to characterize the transferable learning-rate scale through an initialization-side maximal-update law. Within this scope, the multi-path content is not merely the sequential exponent times an $O(1)$ factor: Prop. 3.3 gives explicit lower-order corrections for CNNs, Appendix C derives a residual-block recursion for ResNets, and Appendix D adds the LayerNorm ingredient for Transformers. AM-μP is not ad hoc. Appendix A shows that the arithmetic mean is characterized by permutation invariance, scale equivariance, and merge consistency (with $M(c,\dots,c)=c$), the coarse-graining property needed when depth units are architecture-dependent; by contrast, geometric and harmonic means do not represent the additive second-moment budget correctly under heterogeneity. For the MSE-style simplification, the controlled quantity is the one-step pre-activation update energy, not the loss surrogate itself, and Appendix G explains compatibility with the CE-based experiments. Within the axiomatic setting of Appendix A, the arithmetic mean is unique; exploring whether different desiderata lead to alternative yet useful network-level budgets for multi-path architectures is a natural future direction.
>
> **[5/6] Empirical proxy & direct zero-shot transfer evidence**
>
> The one-epoch optimum is used as a practical calibration of the transferable base learning-rate scale, as intended in Sec. 4.1. To test zero-shot transfer more directly, we ran an explicit experiment on ViT/CIFAR-10: we first calibrate a source LR at source block depth $d_0=12$, then transfer it to all other target depths either unchanged (raw transfer) or rescaled by $(L/L_0)^{-3/2}$ (theory-scaled transfer), where $L$ is effective depth and $L_0=2d_0+2=26$. We compare both against oracle LRs defined by independent grid search at each target depth under the same 3-epoch budget, using epoch-3 average training loss as the selection criterion. In these added experiments, theory-scaled transfer reduces the median log-LR error to oracle from 0.70 decades (raw transfer) to 0.14 decades, and achieves lower epoch-3 training loss than raw transfer on 6/7 target depths. We also ran a preliminary non-vision check (audio), where the best learning rate decreases from $6.31\times 10^{-2}$ at effective depth 6 to $9.03\times 10^{-3}$ at effective depth 18, again showing the same depth-dependent signal.
>
> **[7/9] How to read the fitted spread & current empirical scope**
>
> Our intended empirical claim is a common leading-order depth law, not exact equality of all finite-range fitted slopes to $-1.5$. At practical finite depths, lower-order terms are more visible, particularly for shallower models where fixed stem/head or block-template contributions occupy a larger fraction of the effective depth. Dataset changes can also move finite-range fits through task-dependent constants; in our ViT experiments, CIFAR and ImageNet additionally coincide with different standard architecture choices such as patch size and embedding dimension. This is consistent with the theory: Prop.3.3 gives explicit lower-order corrections for CNNs, while the ResNet/Transformer results are in $\Theta(\cdot)$ form with architecture-template-dependent constants. We are therefore happy to tone down wording such as "universality" / "quantitative agreement" and instead present the result as a shared leading-order law with finite-depth variation. At the same time, the added direct-transfer evidence strengthens the practical claim: the law is not only a post-hoc slope fit but also a predictive calibration rule.

---

> > ### Author Rebuttal · Reviewer_dgob · 2026-04-02
> >
> > Thanks for response. The rebuttal addressed several of my main concerns and improved my evaluation. I could raise my score from 3 to 4.

---

> > > ### Author Response · Authors · 2026-04-08
> > >
> > > Thank you again for the detailed review and for revisiting your evaluation after our rebuttal. We appreciate your careful engagement with the paper and your constructive feedback.

---

### Official Review · Reviewer_1JPy · 2026-03-12

**Soundness:** 3
**Presentation:** 3
**Significance:** 3
**Originality:** 3
**Overall Recommendation:** 4
**Confidence:** 3

**Summary:**

This paper examines how hyperparameter transfer laws as models scale with depth. Similar laws have been proposed and studied wrt width but this is the first treatment to examine similar scaling laws for model depth. In particular, the authors propose a unified framework for the notion of minimal-path and, using this, derive a -1.5 power-law for scaling learning rate with the model depth.

More specifically, the authors look at three types of  non-recurrent, multi-path neural networks including CNNs, ResNets, and Transformers, and introduce a graph-based notion of effective depth. It is this notion of depth for which they derive their scaling law.

**Compliance With Llm Reviewing Policy:**

Affirmed.

**Final Justification:**

Thank you for your clarifying answers. I raised my score in consequence.

**Key Questions For Authors:**

One of the claimed contributions of the authors is the introduction of a graph-based notion of 'effective depth' for deep learning architectures. I have some clarifying questions concerning the authors generalization for irregular or branched architectures that this paper is concerned with (ie. cnns, resnets, and transformers); in particular, with regards to normalization layers. How do normalization layers like Layer Norm or BatchNorm or RMSNorm fit into this definition of effective depth? Or how do Gating MLPs fit into this description as well? If a specific layer does not contain trainable weights but significantly alters the signal variance (like a fixed-scale LayerNorm), should it still be excluded from the depth count? I know LN and BN are considered very briefly in the Ablation Studies section 4.5 and I may have missed something here but this distinction seems crucial for proper measurement of the scaling law constants. How does that affect the predicted $-3/2$ power law?


typo? Line 421 should it be $\alpha = -1.5$ not $\alpha = 1.5$

**Limitations:**

The empirical numbers that are derived to verify the theoretical prediction of -1.5 seems to vary quite a lot. As seen in Table 1, the empirical values range from -1.178 to -1.567. In particular, the value -1.178 is about 20% off the predicted value. This seems like a lot to me but the discrepancy is not really justified, either theoretically or experimentally.

The same is true when looking at the Ablation studies in Section 4.5. The authors look at the robustness of their scaling laws across different activation functions, optimizers, and some normalization layers (batch normalization and Layer norm). Within each category, only one or two variants are considered. It feels quite limited. Also, and more importantly, there again seems to be quite a bit of variance among the ranges of the fitted scaling law constants. But without any attempt at justification. For example, is there some intuition or theoretical motivation why there should be so much variance in the observed experimental scaling laws when changing from SGD to Adam or when using LayerNorm or BatchNorm? See also, my question above regarding effective depth and its role with normalization layers.

**Strengths And Weaknesses:**

**Strengths**
The paper is well-written and well-organized. The authors give a nice, careful, mathematical description for CNNs, ResNets and Transformers. For each of these architectures the authors discuss and provide a clear statement and derivation of their depthwise learning-rate scaling.

This is an important area of research since modern deep learning architectures very costly to train and effective and efficient hyperparameter search is crucial. Among the most critical hyperparameters is the learning rate. It is nice to have a principled approach and mathematically justified method for deriving scaling laws as they pertain to the model depth.

**Weaknesses**
No major weaknesses but I do have some questions and mention some potential limitations below.

---

> ### Author Rebuttal · Authors · 2026-03-31
>
> We appreciate the reviewer's careful engagement with both the technical details and the empirical results.
>
> On effective depth and normalization, our intended definition is not based purely on whether a module has trainable parameters, but on whether it contributes a new sequential update-bearing unit along the minimal path. In our conventions, each plain layer and each residual addition counts as one depth unit, while fixed $O(1)$ internal structure inside a residual branch is absorbed into the branch/block template rather than counted separately. For Transformers, this means each block contributes depth $2$: one unit for the attention residual update and one for the FFN residual update. Under this definition, normalization placement does not itself create an additional depth unit.
>
> For LayerNorm specifically, the current paper already provides formal support. Appendix D derives the LayerNorm Jacobian explicitly and proves an operator-norm bound, showing that for post-norm Transformer units the relevant sensitivity inner products change only by $\Theta(1)$ factors. The pre-norm case follows by the same argument, so the leading $L^{-3/2}$ exponent is unchanged. This is also consistent with the ablation results: Pre-LN and Post-LN give very similar fitted exponents. RMSNorm is not analyzed explicitly in the current draft, but we expect it to behave similarly at leading order. Its Jacobian has the same bounded normalization-plus-rank-one structure and should therefore affect constants rather than the exponent. For BatchNorm, we can add a short analogous argument: BN acts per channel as normalization over the batch-spatial slice, and under $O(1)$ channel variance at initialization its Jacobian has the same projection-normalization structure. It therefore contributes $O(1)$ factors rather than a new depth unit or a different leading exponent, consistent with the ablation curves in Fig.~5, where the BN case still follows a clear power-law fit aligned with the same leading-order depth trend.
>
> Regarding the spread in fitted exponents, our intended claim is a common leading-order depth law rather than exact equality of every finite-range fit to $-1.5$. At finite depths, lower-order terms are more visible, particularly for shallower models where stem/head or block-template contributions account for a non-negligible fraction of the effective depth. Dataset changes can also shift finite-range fits through task-dependent constants and, in our ViT experiments, coincide with different standard architecture choices such as patch size, embedding dimension, and number of heads across CIFAR vs.\ ImageNet. This is consistent with how the theory is stated: Prop.\~3.3 includes explicit lower-order corrections for CNNs, while the ResNet/Transformer results are stated in $\Theta(\cdot)$ form with architecture-template-dependent constants. We therefore interpret Table\~1/Table\~3/Fig.\~5 as finite-depth estimates around the same leading-order asymptotic law. Making these lower-order corrections more explicit, in particular their dependence on architecture choices, normalization, and dataset-dependent constants, is a natural next step for sharpening the finite-depth theory.
>
> For Adam, the empirical result points to a particularly interesting optimizer-aware extension of the framework. Our current maximal-update analysis tracks GD/SGD-style one-step representation-update second moments, whereas Adam's adaptive moment normalization changes the coordinatewise update geometry. Even so, the same depth signal remains clearly visible in Fig.\~5, suggesting that the leading depth dependence is preserved under adaptive preconditioning and motivating a corresponding extension of the theory. Finally, yes: the sign on line 421 should be $\alpha=-1.5$, and we will also fix the same sign typo in the caption of Table\~3.

---

> > ### Author Rebuttal · Reviewer_1JPy · 2026-04-01
> >
> > Thank you for your clarifying answers. I raised my score in consequence.

---

> > > ### Author Response · Authors · 2026-04-08
> > >
> > > Thank you again for the careful reading and for taking the time to revisit the review. We greatly appreciate your thoughtful and constructive feedback.

---

### Official Review · Reviewer_ydEi · 2026-03-15

**Soundness:** 3
**Presentation:** 3
**Significance:** 2
**Originality:** 2
**Overall Recommendation:** 4
**Confidence:** 3

**Summary:**

This paper studies depth-wise scaling of optimal learning rates under the setting of Maximal-Update Parameterizations (muP). By using the arithmetic mean of the original maximal-update criterion used by muP, the authors consider learning rate scaling against the depth of non-recurrent neural network architectures, including MLPs, CNNs, ResNet and Transformers. The authors introduce the concept of effective depth for these architectures and derive a unified power law scaling with exponent -3/2 with respect to the effective depth. Empirically, the authors carry out a set of reasonably comprehensive experiments to validate the scaling exponent for various architectural and data choices.

**Compliance With Llm Reviewing Policy:**

Affirmed.

**Final Justification:**

The authors adequately addressed my concerns and questions. I'd like to maintain a positive score.

**Key Questions For Authors:**

- Can the authors comment on a potential scaling of the optimal learning rate when both depth and width increases?
- Can the authors comment on if the same -3/2 scaling would still hold on pretrained models (so the initializations are not random)?

**Limitations:**

- The authors did not discuss limitations. I think the authors should discuss some potential limitations with respect to my questions in the above.
- The authors did not include a statement of the potential broader impact of their work, which I think is required for ICML. However, based on my reading of the paper, I do not think the subject of this work should have any negative societal impact, so I will not flag it for ethics review.

**Strengths And Weaknesses:**

Strengths:
- The scaling of the optimal learning rate from small to large training setups is an important problem. While the previous line of work mostly focus on the scaling against network width, this work compensates existing results by looking into how learning rates scale across networks of different depths.
- The numerical experiment is reasonably comprehensive and the empirical results are well presented.

Weaknesses:
- The technical contribution is a bit limited to introducing a notation of effective depth and a variant of the maximal-update criterion by taking average over the effective depth. This is a straightforward extension of muP.
- While scaling against network depth is interesting, in practice, usually depth and width are scaled up at the same time. The numerical experiment only considers fixed-width networks. It would be interesting to see how the learning rate scales when both depth and width increases.
- The numerical experiment only considers training the first epoch. Even though this is in line with some previous work, it would be beneficial to see if the same trend also holds in later stages of training.

---

> ### Author Rebuttal · Authors · 2026-03-31
>
> Thank you for the thoughtful and balanced assessment. We are encouraged that you find the problem important and the empirical study reasonably comprehensive.
>
> We will revise the framing to emphasise the paper's main contribution more directly. Our paper develops a unified depth-transfer law for modern non-recurrent multi-path architectures. Beyond introducing an architecture-dependent effective depth, it establishes architecture-specific depth--LR scaling results for CNNs, ResNets, and Transformers, and validates them empirically across model families and datasets. The central technical point is that, once branching and residual aggregation are present, layerwise update statistics are no longer homogeneous, so a transferable depth law must work with an architecture-dependent effective depth together with a network-level maximal-update budget.
>
> On simultaneous scaling in depth and width: our work is meant to complement the original width-transfer guarantees of $\mu$P rather than replace them. In particular, the classical $\mu$P / $\mu$Transfer results already show that many training-critical hyperparameters can be made stable across width under the appropriate parameterization, and our depth law is designed on top of this width-robust setting. This is also reflected in our theory: for CNNs, finite-width terms enter only as lower-order corrections (Prop.~3.3). To address your question empirically, we additionally ran a ViT/CIFAR-10 width-robustness check over three proxy-width settings (small/288, base/384, large/480) with random seed 45. At epoch 1, the fitted slopes are $-1.352$, $-1.360$, and $-1.345$ ($R^2=0.938, 0.980, 0.990$). At epoch 2, the fitted slopes are $-1.531$, $-1.444$, and $-1.647$ ($R^2=0.943, 0.928, 0.961$), all close to the predicted first-order law. At epoch 3, representative wider settings remain very close to theory ($-1.501$ and $-1.550$). We will add these results in tabular form in the revision. The practical joint transfer rule is already given by combining $\mu$P width transfer with our depth law, and these additional results already support that the leading depth exponent remains highly informative across width settings.
>
> | Epoch | Width setting | $\hat{\alpha}$ | $R^{2}$ |
> |------:|:-------------|---------------:|------:|
> | 1 | small (288) | -1.352 | 0.938 |
> | 1 | base (384) | -1.360 | 0.980 |
> | 1 | large (480) | -1.345 | 0.990 |
> | 2 | small (288) | -1.531 | 0.943 |
> | 2 | base (384) | -1.444 | 0.928 |
> | 2 | large (480) | -1.647 | 0.961 |
> | 3 | small (288) | -1.207 | 0.739 |
> | 3 | base (384) | -1.501 | 0.898 |
> | 3 | large (480) | -1.550 | 0.924 |
>
> On the use of the one-epoch optimum: our interpretation is that the theorem identifies the depth-dependent scale for the maximal-update/base learning rate, which can then be used directly or as the target learning rate of a schedule. This is also how Sec.~4.1 motivates our current protocol: the one-epoch optimum is used as an empirical proxy for the maximal-update scale governing early feature updates. In practice, this quantity is valuable because early learning-rate selection strongly influences stability and downstream convergence, while later warmup/decay schedules refine optimization around the chosen base scale rather than replacing it. We further checked epochs 2 and 3 and found that the predicted trend remains informative beyond the first epoch, which is consistent with using the rule as a principled starting point for longer training.
>
> On pretrained checkpoints: pretrained checkpoints are a natural next regime for this framework. The current theorem studies the stabilizing random-initialization regime of Sec.~3.1, where the maximal-update scale is analytically tractable. After pretraining, representation and gradient statistics become data-dependent and are no longer directly tied to these initialization statistics, so extending the same depth law to that regime may require additional calibration or analysis. We will clarify this scope more explicitly, and view pretrained transfer, broader non-recurrent multi-path model families, and schedule-aware transfer beyond the initial calibration stage as promising future directions. We will also add a short Discussion/Future Work paragraph covering these points.

---

> > ### Author Rebuttal · Reviewer_ydEi · 2026-04-02
> >
> > Thank you for the response. I will keep my score of 4.

---

> > > ### Author Response · Authors · 2026-04-08
> > >
> > > Thank you again for the thoughtful and constructive review. We especially appreciate your careful reading and the helpful suggestions you provided.

---

### Decision · Program_Chairs · 2026-04-30

**Decision:**

Accept (regular)

**Comment:**

The reviewers converged to a weak recommendation for acceptance, and I agree with this assessment. The paper addresses a practically important problem (depth-wise learning rate transfer) and provides a unified framework spanning CNNs, ResNets, and Transformers through a graph-based effective depth notion. The theoretical development is coherent, and the empirical validation across architectures and datasets is reasonably comprehensive. The rebuttal was thorough, leading part of the committee to raise their scores. That said, I share some of the reviewers' reservations, particularly about the contribution being more of a careful synthesis and extension than a fundamentally new discovery. The title and framing should be narrowed to match the actual contribution (learning-rate scaling, not general hyperparameter transfer). I encourage the authors to incorporate the promised revisions into the final version.